# Formulation of Dosage Forms with Proton Pump Inhibitors: State of the Art, Challenges and Future Perspectives

**DOI:** 10.3390/pharmaceutics14102043

**Published:** 2022-09-25

**Authors:** Justyna Srebro, Witold Brniak, Aleksander Mendyk

**Affiliations:** Department of Pharmaceutical Technology and Biopharmaceutics, Jagiellonian University Medical College, Medyczna 9, 30-688 Krakow, Poland

**Keywords:** proton pump inhibitors, delayed-release tablets, enteric coating, Eudragit, omeprazole, pantoprazole, lansoprazole

## Abstract

Since their introduction to pharmacotherapy, proton pump inhibitors (PPIs) have been widely used in the treatment of numerous diseases manifested by excessive secretion of gastric acid. Despite that, there are still unmet needs regarding their availability for patients of all age groups. Their poor stability hinders the development of formulations in which dose can be easily adjusted. The aim of this review is to describe the discovery and development of PPIs, discuss formulation issues, and present the contemporary solutions, possibilities, and challenges in formulation development. The review outlines the physicochemical characteristics of PPIs, connects them with pharmacokinetic and pharmacodynamic properties, and describes the stability of PPIs, including the identification of the most important factors affecting them. Moreover, the possibilities for qualitative and quantitative analysis of PPIs are briefly depicted. This review also characterizes commercial preparations with PPIs available in the US and EU. The major part of the review is focused on the presentation of the state of the art in the development of novel formulations with PPIs covering various approaches employed in this process: nanoparticles, microparticles, minitablets, pellets, bilayer, floating, and mucoadhesive tablets, as well as parenteral, transdermal, and rectal preparations. It also anticipates further possibilities in the development of PPIs dosage forms. It is especially addressed to the researchers developing new formulations containing PPIs, since it covers the most important formulary issues that need to be considered before a decision on the selection of the formula is made. It may help in avoiding unnecessary efforts in this process and choosing the best approach. The review also presents an up-to-date database of publications focused on the pharmaceutical technology of formulations with PPIs.

## 1. Introduction

Proton Pump Inhibitors, also known as PPIs, belong to a group of antisecretory drugs [1,2]. Along with histamine H_2_-receptor antagonists and potassium-competitive acid blockers (PCAB), they are used in the treatment of gastroesophageal reflux disease (GERD) and other disorders characterized by excessive secretion of gastric acid [3,4,5,6]. According to the IQVIA report on *Medicine Spending and Affordability in the United States*, omeprazole and pantoprazole were included in the list of the 20 most commonly prescribed medicines in 2020 [7]. Furthermore, in the United Kingdom, omeprazole was the leading antisecretory drug, with over 35 million items dispensed in 2021 [8]. Despite their wide use in pharmacotherapy, there are still unmet needs in the availability of appropriate dosage forms for patients of all age groups. Therefore, the development of novel improved medicinal products with PPIs is fully reasonable and should be supported. The wider availability of more sophisticated formulations that are easy to prepare and to be administered, such as minitablets, orodispersible tablets and films, especially those containing functional micro- or nanoparticles is necessary. Standard enteric-coated tablets or capsules are not suitable for all patients, leading to common problems, such as the need for dose adjustment, crushing or grinding of such forms. It also negatively influences the compliance of patients and their medication adherence. Currently developed formulations are mostly intended to overcome these difficulties and increase the effectiveness and safety of therapy with PPIs.

The aim of this review is to describe the historical background of proton pump inhibitors discovery and development, discuss formulation issues related to this family of drugs, present contemporary solutions, discuss further possibilities, and challenges in formulation development.

The physicochemical characteristic of PPIs is briefly explained, especially in relation to the pharmacokinetic and pharmacodynamic properties of PPIs, their stability, and methods of qualitative or quantitative analysis. We analyzed the composition and kind of formulation in commercially available preparations with PPIs. The most important part of this review describes the numerous approaches to formulate different dosage forms with PPIs: nanoparticles, microparticles, minitablets, pellets, bilayer tablets, gastroretentive tablets, and mucoadhesive tablets, as well as dosage forms administered with non-oral routes, such as parenteral, transdermal, and rectal preparations.

A brief description on the methodology used to prepare this review was described in the Appendix A.

### 1.1. Historical Background

The first antisecretory drug registered for the treatment of peptic ulcers and GERD was cimetidine, the competitive H_2_-receptor antagonist, launched in 1976 [9,10] (Figure 1). The mechanism of its action was based on blocking paracrine stimulation of parietal cells [2]. Soon, it was found that H_2_-receptor blockers were effective in reducing basal gastric acid secretion, but only partially inhibited postprandial secretion in parietal cells [3,4]. The same year, it was recognized that the ATP-dependent H^+^/K^+^ proton pump is the essential component of the gastric acid secretion process in parietal cells. Blocking the proton pump has been concluded to be more effective than inhibiting receptors responsible for paracrine stimulation of acid secretion [11]. Two years later, in 1978 it was discovered that recently developed benzimidazole derivatives with a potent antisecretory effect—timoprazole and picoprazole—directly inhibitH^+^/K^+^-ATPase in parietal cells [9,12]. However, further studies on timoprazole and picoprazole revealed their toxicity in the thyroid gland and blood vessels, respectively, which led to a necessary optimization of the benzimidazole structure. After the addition of substituents to the pyridine ring of picoprazole, which increased its pKa value, an omeprazole was finally developed [9].

The expected superiority of omeprazole over H_2_-receptor antagonists in the treatment of GERD, gastric, and duodenal ulcers was proved in clinical trials, and it was launched on the European market in 1988 by a company currently known as Astra Zeneca. Two years later, the first medicinal product containing omeprazole was introduced to the United States under the brand name Prilosec [9,11].

Subsequently, other proton pump inhibitors were developed and launched on the market, including lansoprazole, pantoprazole, rabeprazole, esomeprazole and dexlansoprazole [11,19,20]. The next advancement was the introduction of a new generation of long-term-acting PPIs, characterized by extended plasma half-life. The only substance belonging to this group that has already been marketed is ilaprazole. Other drugs, currently in clinical trials, include tenatoprazole, AGN201904-Z (Durasec™), azeloprazole, anaprazole, and DLBS-2411 (Redacid^®^). The latter is a plant-derived therapeutic substance that inhibits the expression of the proton pump messenger RNA in parietal cells [21]. AGN201904-Z is an omeprazole prodrug that is more stable in acidic media compared to the native molecule [21,22,23]. On the contrary, tenatoprazole (benatoprazole) has a chemical structure based on the imidazopyridine ring, rather than the benzimidazole moiety [24,25].

Following the successful introduction of PPIs into the market, in the 1980s a new class of antisecretory drugs, i.e., potassium-competitive acid blockers (PCAB) was developed. PCABs were characterized by a mechanism of action similar to that of PPIs, but unlike the proton pump inhibitors, they reversibly bond to the H^+^/K^+^-ATPase. Currently marketed PCABs are revaprazan, vonoprazan, and tegoprazan [5,6].

Another group of drugs known for their antisecretory activity are cholecystokinin (CCK) receptor antagonists. Cholecystokinin is a peptide hormone that regulates gastric acid secretion by binding to CCK_2_-receptors in the gastric mucosa [13]. The first nonspecific cholecystokinin receptor antagonist used in the therapy of peptic ulcers was proglumide. However, it is no longer marketed in the US and Europe because it was replaced by newer, more potent antisecretory drugs. The other drugs which belong to this group are lorglumide and devazepide. They are specific CCK_2_ antagonists; however, they have not yet been marketed and are still only the subject of scientific studies [13,14].

### 1.2. Pharmacokinetics and Pharmacodynamics

Proton pump inhibitors have been found to be very effective in suppressing gastric acid secretion. They share the same mechanism of action, although there are slight differences in their chemical structure [20]. PPIs inhibit the activity of the enzyme H+/K+-ATPase, also named gastric proton pump, located in the parietal cells of the stomach. The function of the proton pump is the secretion of acid into the lumen of the stomach, during the basal secretion or in response to stimuli, such as hormones, peptides, or neurotransmitters [12].

Proton pump inhibitors are inactive compounds (often simply but incorrectly called ‘prodrugs’), which require activation in the low pH of parietal cells, to suppress the activity of the proton pump [19]. To avoid premature activation in the stomach after oral administration, they must be protected from gastric acid, e.g., with enteric coating. PPIs are absorbed in the duodenum. They are weak acids, with pKa values ranging from about 4.0 (omeprazole) to 5.0 (rabeprazole) (Table 1). In a weakly acidic environment of the duodenum (pH ≈ 5.6), their form becomes unprotonated and therefore can be effectively absorbed through enterocytes. After absorption, PPIs are transported through the blood system to the canaliculi of parietal cells, where they accumulate. In the acidic environment of canaliculi, the inactive forms are protonated into cyclic sulfonamides or sulfonic acids [19]. PPIs with higher pKa values are characterized by a faster onset of the protonation process and greater accumulation in parietal cells [2,19,26,27]. The biotransformed compounds covalently bond to cysteine residues in the α-subunit of the proton pump, forming disulfide bridges. In particular, covalent bonding with Cys813 is considered to suppress the activity of the proton pump [2,9].

It was assumed that the bonding of the drug to the enzyme is irreversible, therefore, the restoration of its activity depends on the biosynthesis of the new proton pumps in parietal cells. However, it was observed that the enzyme can be reactivated by endogenic glutathione [2,9,20,27]. Its restoring activity is associated with the location of cysteine residues, which create the additional disulfide bridges, specific for each PPI. Because of this, differences in the half-life of PPIs inhibitory complexes with enzymes can be noted. For instance, pantoprazole is bonded to the 813 and 822 cysteine residues, while the omeprazole binding sites are cysteine 813 and 892. The half-time of their complexes with the enzyme is 46 h and 27 h, respectively [2,19].

PPIs are capable of forming stable inhibitory complexes only with active proton pumps, which are activated in response to a meal, triggering a cascade of stimuli, such as hormones, peptides, and other transmitters [20]. However, due to the relatively short half-life of PPIs (0.5–1.5 h, except tenatoprazole), after oral administration, about 70% of active proton pumps can be effectively inhibited [2,19,20,26]. Therefore, to obtain the highest efficacy of H+/K+-ATPases inhibition, PPIs should be administered 30–60 min before breakfast or other substantial meals [2]. According to a recent review by Weisner et al. [18] some formulations of PPIs, such as delayed-release tablets, can be taken independently of food intake. Studies indicate that pantoprazole, rabeprazole, and dexlansoprazole are generally less affected by concomitant food intake than omeprazole, lansoprazole, or esomeprazole [18]. After 2–3 days of therapy, the steady state is achieved, in which 66% of gastric acid secretion is inhibited, and thus the relief of symptoms is noticeable. Administration of PPIs twice daily increases the maximum secretory inhibition to 80% [19]. However, additional administration of PPIs does not bring significant relief in nocturnal acid breakthroughs, because of their short half-life. The exception for that is tenatoprazole, with a half-life of up to 9 h [19]. This drug efficiently suppresses nighttime acid secretion, even after stopping treatment [24]. However, its inhibitory activity is not satisfactory during the day [19].

To describe the efficacy of the inhibitory activity of PPIs, both the maximum concentration (C_max_) and the area under concentration versus time curve (AUC) are used. There is a limit of the drug plasma concentration, above which the proton pumps are efficiently inhibited. The longer the plasma level is kept above this threshold, the more efficient the drug is. Therefore, simple C_max_ values for PPIs are not sufficient to describe the value of gastric acid suppression [26,27]. PPIs are characterized by high bioavailability and a high level of protein binding. They are metabolized in the liver, mostly by the CYP219 isoenzyme. Their metabolism is stereoselective—for instance, the R-enantiomer of omeprazole is metabolized rapidly, whereas the S-enantiomer is cleared much more slowly. Furthermore, PPIs metabolism is correlated with CYP219 isoenzyme polymorphism. Slow metabolizers (mostly in the Asian population) have higher plasma concentrations and a longer half-life of the drug than fast metabolizers. However, the PPIs that are metabolized in nonenzymatic reactions, like rabeprazole, are less likely to be prone to isoenzyme polymorphism and therefore have a drug–drug interaction potential smaller than that of others. Proton pump inhibitors are excreted in urine and feces [2,20,27].

### 1.3. Medical Uses

PPIs can be administered orally or intravenously. Due to their antisecretory action, they are used in the treatment of gastrointestinal diseases, such as:Gastroesophageal Reflux Disease (GERD),Functional dyspepsia,Erosive/Non-erosive Esophagitis,Gastric and duodenal ulcers,Helicobacter pylori infections (combination therapy),Hypersecretory syndromes (e.g., Zollinger-Ellison syndrome),and in the prevention of NSAID-induced gastroduodenal ulcers.

Proton pump inhibitors are characterized by almost similar antisecretory activity. The choice of PPI depends on the expected clinical effect defined by the specific pharmacokinetic properties and the dosage form of the drug [20,35].

## 2. The Most Important Issues to Be Considered in the Formulation of Medicinal Products with PPIs

The formulation of medicinal products containing proton pump inhibitors is a challenging process, mainly due to their low water solubility and stability problems [36]. The current section describes the physicochemical properties of PPIs. The most important factors affecting stability are depicted here. At the end of this section we briefly described the challenges associated with the qualitative and quantitative analysis of PPIs and provided the most relevant examples of the analytical methods.

### 2.1. Physicochemical Properties of PPIs

The molecular structure of all proton pump inhibitors (beside tenatoprazole) is based on the 2-pyridylmethylsulfinylbenzimidazole moiety (Figure 2). It can exist in several states of protonation depending on the pH of the solution. Therefore, they can be characterized by two or even three pKa values. The first value of pKa ranging from 3.55 to 4.77 is associated with the acceptance of protons in the nitrogen atom of pyridine (marked red in Figure 2) in an acidic environment and the second results from the dissociation of a proton from the benzimidazole ring (marked green in Figure 2) in presence of alkalis [37]. The pKa_2_ values reported in the literature range from 9.15 for pantoprazole to 10.10 for ilaprazole. Some articles also described the third pKa value associated with the protonation of the nitrogen atom at the third position of the benzimidazole ring (marked pink in Figure 2) [38]. Its value ranges from −0.11 to 0.79, which is much below the pH value in which PPIs form stable compounds. Yang et al. [39] described these values as an estimate, based only on kinetic potentiometric studies of decomposition without real experimental confirmation.

Another important aspect of the chemical structure of proton pump inhibitors is their chirality. PPIs possess the asymmetric sulfur molecule in the sulfinyl moiety, which binds pyridine with the benzimidazole group or, in the case of tenatoprazole, with the imidazopyridine core. Therefore, all PPIs may exist in the forms of S- or R-enantiomers as well as racemates. Both forms are equally pharmacodynamically active at the target site and do not make a difference in adverse effects. However, the level of plasma protein binding and metabolism by cytochrome P450 shows stereoselectivity, leading to variability in the pharmacokinetics of pure enantiomers [40,41]. Esomeprazole, being an S (−) isomer of omeprazole, undergoes a metabolic clearance three times lower than that of the R-enantiomer. Its metabolism shows a significantly lower dependency on CYP2C19 isoform, leading to a much lower difference in pharmacokinetics between poor and extensive metabolizers, making S-isomer pharmacotherapy more predictable [41]. Some advantages over opposite forms or racemates were also proved in the case of dexlansoprazole, which is the R (+) enantiomer of lansoprazole and dexrabeprazole, which is the R (+) isomer of rabeprazole. This caused their introduction to the market in the form of single enantiomers. In the case of other PPIs, such a difference was not beneficial enough to market them separately [40,41].

PPIs are very slightly soluble in water but dissolve easily in alkali solutions or ethanol. In the form of sodium or magnesium salts (e.g., omeprazole sodium) are freely soluble in water and ethanol and therefore are used in intravenous administration. Proton pump inhibitors are lipophilic drugs with logP values in the range of 1.6 to 2.8 [28]. Omeprazole and lansoprazole, as well as their stereoisomers, belong to class II of the Biopharmaceutical Classification System, due to their poor solubility in water and high permeability through cell membranes [29,30,31,32]. On the other hand, pantoprazole and rabeprazole were classified as provisional BCS class III [33,34].

### 2.2. Stability of Proton Pump Inhibitors

Proton pump inhibitors are sensitive to the acidic environment, light, temperature, oxidative conditions, and the presence of other salts. As mentioned before, proton pump inhibitors are inactive compounds that have to be activated by the protonation in the acidic environment of parietal cells. Therefore, after oral administration, PPIs must be protected against premature activation in the stomach. They are relatively stable at pH = 7.0, but quickly decompose in acidic solutions. The degradation of omeprazole at low pH occurs within 24 h while it is most stable at pH = 11.0 [42]. It was found that in solutions with pH above 7.8 the degradation of this PPI followed the first-order kinetics.

### 2.3. Stability in Solutions

DellaGreca et al. [43] investigated the degradation products of omeprazole and lansoprazole in aqueous solutions using NMR spectroscopy. Hydrolysis reaction was observed in water solutions and solutions adjusted to a pH of 4.0. At pH of 7.0 and 9.0 lansoprazole and omeprazole remained unchanged for 43 and 72 h, respectively. For both drugs, the main hydrolysis products were benzimidazolones, sulfides, and the red residue identified as a very labile mixture of degradation products, impossible to separate. Exposition of the water solution and solutions adjusted to a pH = 7.0 to solar light accelerated the degradation of the PPIs. In this case, dianilines, pyridines, and benzimidazoles were found among the degradation products [43].

Studies on the stability of rabeprazole revealed that it was highly unstable in 0.1 M hydrochloric acid solution, as well as in a 30% solution of hydrogen peroxide, in which it decomposed within 60 min [44].

Mahadik et al. [45] studied the stability of tenatoprazole under various stress conditions. Hydrolytic decomposition of the drug substance was measured in 0.1 M hydrochloric acid. After 30 min of study, 40% of tenatoprazole remained unchanged. At a lower concentration of the acid solution (0.01 M), complete degradation of the substance occurred after 4 h. In a basic environment of 1 M sodium hydroxide at 80 °C, approximately 20% of tenatoprazole decomposed after 4 h of incubation. The drug substance was also unstable under oxidative conditions of 30% hydrogen peroxide solution, in which 60% of the drug degraded in 1 h. After exposure to solar light for 1 h, more than 20% of tenatoprazole has been decomposed. Despite that, the substance in its solid form remained stable for 2 months exposed to dry heat (50 °C) [45]. In another research, it was found that omeprazole solutions in the presence of acids change their color to yellow, dark red, brown or purple, whereas pharmaceutical formulations containing PPIs were unstable under heat and moisture conditions, changing their color from brown to dark brown [46].

### 2.4. Influence of Temperature

Several research studies evaluated the effect of storage temperature on the stability of omeprazole, which has a significant impact on extemporary compounded medicines, which are usually prepared for pediatric patients. In one of them, the 2 mg/mL omeprazole solution in 8.4% sodium bicarbonate was kept at −20 °C, 5 °C, and 24 °C for 30 days. The liquid stored at room temperature gradually changed color to brown, indicating the appearance of degradation products. At the end of the study, the omeprazole content in the sample was 84.2% of the initial concentration. The solutions stored at low temperatures remained stable for the research period [47]. It was found in another study that the half-life of omeprazole in the solution at pH 7.5 at a temperature of 4 °C was 125 days, while at a temperature of 40 °C it was only 42 h [48].

### 2.5. Influence of Salts

The stability may be affected not only by the pH value but also by the presence of other ions in the solution. In one of the studies on the stability of omeprazole, pantoprazole and lansoprazole, it was found, that the most stable compound in the pH range of 4.0 to 7.0 was pantoprazole, whereas the least stable was lansoprazole. The degradation of PPIs was related to the increasing concentration of hydrogen ions and salts in the solutions. The lowest stability of the investigated PPIs was observed in the 0.5 M citric, phosphate, and acetate buffer solutions (at pH = 6), as well as in the trisodium citrate solution (0.025 M and 0.25 M). The highest stability was achieved in the water and sodium chloride solutions (0.05 M and 0.5 M) in the entire range of measured pH values [49]. The half-life of pantoprazole sodium in phosphate-buffered solution maintained at pH = 7.4 was found to be approximately 124 h [50].

### 2.6. Influence of Light

Dhurke et al. [51] investigated the effect of UVA, UVC, and solar light on the degradation of pure and microencapsulated pantoprazole. After seven days of exposure to pure pantoprazole to UVC radiation (254 nm), its degradation rate was 38.09%, while in UVA radiation conditions (366 nm) 35.11%. What is more, the half-life of pure pantoprazole affected by solar light was found to be 8.6 days. The process of microencapsulation with the Eudragit S significantly improved the stability of pantoprazole during exposure to all types of radiation investigated, because the physical barrier for light was formed [51].

The impact of UVC radiation on pantoprazole was also measured by Raffin et al. [52]. Pantoprazole methanolic solutions were exposed to radiation at 254 nm. It has been reported that 98.8% of API degraded within 120 min of the study. The stability of pantoprazole powder was higher than that of the methanolic solution, i.e., 27% of the drug remained stable after 10 days of exposure to UVC radiation. Similarly to the previously mentioned study of Dhurke, microencapsulation with Eudragit S100 increased the stability of pantoprazole. After 10 days of exposure to UVC radiation, 55% of the API remained unchanged [52].

Garcia et al. [44] investigated the stability of rabeprazole methanolic solutions (800 µL/mL) exposed to UVC radiation. It was observed that 88% of the substance decomposed in 30 min, characterized by zero-order kinetics. It was found that the two main photodegradation products of rabeprazole were benzimidazole and benzimidazolone. The same experimental conditions were applied to the crushed and solid tablets containing rabeprazole—the degradation products of API could be observed after 10 and 50 days, respectively. The higher stability of rabeprazole in solid form than in solution was explained by the presence of excipients and the smaller surface area exposed to radiation in the case of tablets. Furthermore, tablets containing rabeprazole were found to remain unchanged for 4 days at a temperature of 80 °C [44].

### 2.7. Interaction of Enteric Polymers with PPIs and Its Effect on the Stability

One of the most challenging aspects of the PPIs’ pharmaceutical development is their instability under acidic conditions. To avoid premature release and degradation of API in the stomach, enteric-coating polymers are used. The gastro-resistant coating provides delayed, site-specific, and pH-dependent release of the drug substance in the small intestine. Enteric polymers are composed of long chains of organic monomers with free carboxylic residues. Their pKa values range between 4 and 6. Acidic moieties of enteric polymers undergo ionization and dissolve rapidly in the small intestine, but remain unionized in an acidic environment [36,46,53]. Examples of polymers commonly used for enteric coating are polymethacrylates (Eudragit^®^ L, Eudragit^®^ S), cellulose derivatives (cellulose acetate phthalate, CAP), polyvinyl derivatives (Opadry^®^ Enteric), resins (AquaGold^®^ shellac), and starch derivatives (Aqua-Zein^®^) [36]. Unfortunately, the presence of free acidic moieties in the tablet coat or in tablet mass can affect the stability of PPIs. A series of experiments conducted by Riedel et al. [54,55] provided data on the molecular interactions between omeprazole and enteric coating polymers [54,55].

They evaluated with RP-HPLC the degradation of omeprazole in methanolic and aqueous dispersions of enteric coating polymers such as hydroxypropylmethylcellulose phthalate (HP-55), hydroxypropylmethylcellulose acetate succinate (HPMCAS-HF/-LF), shellac, cellulose acetate phalate (CAP), Eudragit L-100 and S-100. The degradation rate of omeprazole was significantly lower in the organic solution. Its stability was correlated with the number of acid moieties in the polymer structure, as well as the presence of acidic impurities. For instance, the decrease in the peak area of omeprazole was more pronounced in the Eudragit L 100 solution (27%) than in HPMCAS-LF (16.7%), which was correlated with the number of acidic moieties in the polymer structure, i.e., 46–50% and 14–18% respectively. The highest rate of drug degradation was observed in the methanolic solutions of CAP and HP-55 (79% and 35%). Although these polymers contain a smaller amount of acid moieties than Eudragit L 100, the presence of impurities, such as phthalic acid, noticeably influenced the results of the analysis. It was concluded that the reaction of omeprazole degradation in organic solutions followed first-order kinetics. The decomposition of omeprazole in aqueous dispersions was influenced by the low pH and the pKa values of the polymers. During the 3 h analysis, omeprazole was almost completely decomposed in polymeric dispersions. No differences in the amount of degradation products were observed between Eudragit L 100 and CAP. The degradation of omeprazole was the least pronounced in shellac solutions, both aqueous and methanolic (a decrease in the peak area 74% and 3%, respectively). This can be explained by the less acidic properties of shellac (the highest pKa value) among the investigated polymers [54].

In the subsequent study, Riedel et al. [55] investigated omeprazole decomposition in organic and aqueous solutions of different polymers including Eudragit L 100, Eudragit S 100, HPMCAS-HF, HPMCAS-LF, Aquateric (cellulose acetate phthalate), HP-55 and shellac, the ethanolic acetic acid solution, and Eudragit RS 100. At the beginning of the study, no discoloration was observed in the investigated polymer solutions. However, after 3 h, the solutions changed colors to purple (Eudragit S 100/L 100, HPMAS, CAP), yellow (HP-55), and red (shellac), indicating degradation of the drug substance. On the other hand, omeprazole was compatible with the ethanolic solution of Eudragit RS (cationic polymethacrylate) because it does not contain acid groups in its structure. In the presence of monomeric acetic acid, the degradation of omeprazole was more exacerbated than in the presence of polymeric acids Eudragit S 100 and L 100.

Stroyer et al. [46] evaluated the influence of enteric polymers on powdered omeprazole. The coating polymers used for the study were Eudragit L 100 (and separately its mixture with sodium citrate), Eudragit RS PO, shellac, hypromellose acetate succinate (HPMCAS-HF), and hydroxypropylmethylcellulose phthalate (HP-55). The results of the study showed that, in the case of the powder blends stored at room temperature, the presence of degradation products has not been detected. However, in samples kept under accelerated storage conditions (40 °C/75%RH), white powder mixtures (except shellac) changed colors from pink to brown depending on the number of degradation products in the sample. The highest amount of decomposition products was found in the omeprazole mix with HP-55 (10%), shellac (8%), HPMCAS-HF (6%) and Eudragit L 100 with sodium citrate (3%). No correlation was found between the amount of decomposition products and the acidity of the coating substance. The mechanism of this is probably due to the fact that some polymers, such as methacrylates, have the ability to bind water available for chemical reactions with the drug substance. Therefore, despite the presence of acidic moieties in the structure of the polymer, API remains relatively stable in contact with the coating substance. In this context, the degradation of omeprazole mixed with HP-55 probably resulted from the presence of acidic impurities, i.e., water-soluble phthalic acid, that entered the reaction with API. Decomposition products were also detected in the shellac mixture under accelerated storage conditions, which was explained by the shellac melting that resulted in a decrease in its pH value (5.1) [46].

To avoid the risk of interaction between the enteric coating layer and the drug substance, an inert polymer can be applied to create a separating layer on the tablet or granule core. For this purpose, cellulose derivatives (HPC, HPMC), sucrose, polyethylene glycol, and povidone are generally used. A separating coat can also be formed with alkaline substances, such as sodium salts of weak inorganic or organic acids. Moreover, it was described that a pH buffering coat can be created in situ. In this case, a separation layer is formed in the reaction of the alkaline base material with an acidic enteric coating material [46].

### 2.8. Analytical Methods for PPIs Determination

The quantitative and qualitative analysis of PPIs with a simple method such as UV-Vis spectrophotometry may be a bit challenging, even though all of them have distinct peaks with maximum absorption in the range 280–302 nm. Their low stability in solutions depending on pH, susceptibility to light, and oxidation lead to the formation of multiple degradation products. Some of them show absorbance in similar range as initial compound, thus, employment of separation techniques may be necessary. Therefore, besides spectrophotometry, also high performance and ultra-high performance liquid chromatography (HPLC and UPLC) are commonly used to identify PPIs and their degradation products or to measure the concentration of PPIs in pharmaceutical formulations or biological samples [56,57] (Table 2).

There are also multiple examples of the use of chemometric techniques for spectroscopy-based analysis of PPIs [51,52]. Wahbi et al. [58] applied compensative method, derivative, orthogonal function, and difference spectrophotometry techniques for the determination of omeprazole, lansoprazole, and pantoprazole concentrations in pharmaceutical gastro-resistant forms. The wavelengths used for the analysis were 306.2 nm, 292.4 nm, and 295.4 nm for omeprazole, lansoprazole, and pantoprazole, respectively. The difference spectrophotometric method was found to be suitable for stability-indicating assays, due to the lack of an interference impact of the degradation products of PPIs [58].

El-Sherif et al. [59] described an HPLC method for the determination of the content of lansoprazole, omeprazole, and pantoprazole in the presence of their acid degradation products. The method was suitable for the determination of drug substances in bulk as well as in dosage forms. The analysis was carried out using a Waters Nova-Pak C_18_ 60 Å column with the mobile phase composed of 0.05 N potassium dihydrogen phosphate, methanol, and acetonitrile in a ratio of 5:3:2 (*v*/*v*/*v*). The retention times were found to be 2.10 min for omeprazole, 3.34 min for lansoprazole and 4.54 min for pantoprazole. The method was linear in the following PPI concentrations: 2–20 mg/mL (lansoprazole), 2–36 mg/mL (omeprazole) and 0.5–20 mg/mL (pantoprazole). Additionally, due to its selectivity, the method allowed the separation of PPIs in the presence of 7 degradation products [59].

Another HPLC method of determination of omeprazole, lansoprazole, pantoprazole, and rabeprazole was designed for their determination in human plasma for pharmacokinetic studies. Analysis was carried out using a Zorbax C_8_ column with a mobile phase composed of 0.1% triethylamine (pH = 6.0) and acetonitrile in a ratio of 72:28 (*v*/*v*). The analysis run time was 11 min, and the method was linear in the range of 20.61–1999.79 ng/mL [60].

There are also examples of using other methods for the analysis of PPIs, among them with simultaneous determination of several compounds, e.g., differential pulse polarography, square-wave voltammetry, LC-MS/MS, or TLC (Table 2).

Elkady et al. [61] developed a method for the simultaneous separation of rabeprazole, pantoprazole, lansoprazole, and esomeprazole from human plasma by LC-MS/MS. Chromatographic separation of drug substances was carried out on an RP-C18 column using 10 mM ammonium formate:acetonitrile:methanol in a 20:40:40 ratio (*v*/*v*/*v*) as mobile phase. The retention times of the PPIs ranged from 2.77 for lansoprazole to 3.08 min for pantoprazole. Furthermore, escitalopram was used as an internal standard, for which the retention time was 2.09 min. The isocratic elution was carried out at 40 °C, with a flow rate of 0.8 mL/min and an injection volume of 5 µL. For tandem mass spectroscopy, positive mode electrospray was used as the ionization source. The mass-to-charge ratio for the detection of lansoprazole was *m*/*z* 370.1→252, for esomeprazole *m*/*z* 346.2→198.1, for rabeprazole *m*/*z* 360.1→242.1 and *m*/*z* 384.2→200.2 for pantoprazole. The developed method provided linearity in the concentration range of 20–5000 ng/mL [61]. Chunduri et al. [62] used the UPLC-MS/MS method for the simultaneous determination of rabeprazole, esomeprazole and levosulpiride in human plasma. Drug substances were separated using gradient elution, where the mobile phase was 2 mM ammonium formate and acetonitrile at a flow rate of 0.5 mL/min. Lansoprazole was used as an internal standard. In MS/MS analysis of rabeprazole and esomeprazole, ions were detected at *m*/*z* 360.1→242.1 and *m*/*z* 346.1→198.1, respectively. The authors proved the linearity of the method in the concentration range of 0.1 to 2000 ng/mL for each substance tested [62].

For the quantification of dexlansoprazole in bulk and in pharmaceutical preparations, Bora et al. [63] proposed a LC-tandem mass spectroscopy method using omeprazole as an internal standard. Substance separation was carried out on a Zorbax SB C18 column using an isocratic flow of the mobile phase composed of 0.5 mM ammonium acetate (pH = 3.5) and acetonitrile in a ratio of 30:70 (*v*/*v*). A flow rate of 0.5 mL/min and an injection volume of 10 µL were used. The *m*/*z* ratio observed by mass spectroscopy for dexlansoprazole was 255→237.1 and 195→138.1 for omeprazole. The method was shown to be linear in the concentration range of 0.5–3000 ng/mL for the tested substance [63].

To determine pantoprazole in human plasma, Li et al. [64] developed an LC-MS/MS method, which was used to study the pharmacokinetics and bioequivalence of enteral-coated pantoprazole capsules. As an internal standard, omeprazole was used. The separation of components was carried out in a reverse phase system at 40 °C with a flow rate of 0.3 mL/min. The mobile phase was methanol:water in a 60:40 (*v*/*v*) ratio with the addition of 1% of ammonium acetate. Tandem mass spectroscopy was performed by positive ion electrospray ionization. Pantoprazole ions detection was carried out at *m*/*z* 384.1→200.0. The method was shown to be linear in the range of 5–5000 ng/mL [64].

In other studies, the LC-MS/MS method was used for the simultaneous determination of esomeprazole and naproxen in human plasma [65], as well as pantoprazole and amitriptyline in rabbit plasma [66].

Another aspect of the PPIs analysis is the chiral separation of a particular enantiomer. This issue has already been briefly described in the literature [67,68]. One example is the application of liquid chromatography and mass spectrometry (LC-MS/MS) for the simultaneous determination of omeprazole, pantoprazole, lansoprazole, rabeprazole, and ilaprazole enantiomers. The analysis was conducted on human plasma samples and no significant interference from the matrix was observed. The developed method was also suitable in combination with equilibrium dialysis for the studies of PPIs enantiomers ratio binding with plasma proteins. The linearity of the method was determined in the range of 1.25 to 2500 ng/mL for the investigated enantiomers [68].

A more detailed review of the analytical chromatographic and electrophoretic methods used for the determination and quantification in bulk, pharmaceutical formulations, and biological fluids was provided by Joshi et al. [56] and by El-Kommos et al. [69].
pharmaceutics-14-02043-t002_Table 2Table 2Examples of analytical methods for PPIs determination.PPIAnalytical MethodDetailsReferenceomeprazoleUV-Vis spectrophotometryformation of colored species in reaction with 3-methyl-2-benzothiazolinone hydrazone (MBTH)[70]UV-Visspectrophotometry, 2nd derivative methodlinearity in the range of 0.2–40.0 µg/mL[71]differential pulse polarographystatic mercury electrode;linearity in the range of 0.2–20 µmol/L[72]RP-HPLCUV-Vis detection at λ = 280 nm;mobile phase phosphate buffer (pH = 7.4):acetonitrile (70:30);linearity in the range of 10.0–30.0 µg/mL[73]esomeprazoleRP-HPLCUV-Vis detection at λ = 300 nm;mobile phase acetonitrile:methanol (50:50);linearity in the range of 5.0–25.0 µg/mL[74]lansoprazolesquare-wave voltammetryhanging mercury drop electrode (HMDE);pH of investigated solutions 2.0–11.0;linearity in the range of 1.0 × 10^−9^–5.0 × 10^−8^ M[75]RP-HPLCUV-Vis detection at λ = 284 nm;mobile phase methanol:water (80:20);linearity in the range of 50.0–30.0 μg/mL[76]LC-MS/MSmobile phase water:acetonitrile with 0.1% formic acid (60:40);IT-TOF detection; linearity in the range of 5.0–25.0 µg/mL[77]pantoprazoleRP-HPLCUV-Vis detection at λ = 289 nm;mobile phase potassium dihydrogen solution:acetonitrile (70:30);linearity in the range of 20.0–200.0 µg/mL[78]LC-ESI-MS/MSLC mobile phase acetonitrile:water:methanol (57:25:18) with addition of 10 mmol/L acetic acid and 20 mmol/L ammonium acetate; transition *m*/*z* 383.8→199.6; linearity in the range of 5–5000 ng/mL[79]Chiral LC-MS/MSLC mobile phase 10 mM ammonium acetate solution containing 0.1% acetic acid:acetonitrile (28:72); transition *m*/*z* 384.1→200;linearity in the range of 5–10,000 ng/mL[80]rabeprazoleLC-ESI-MS/MSLC mobile phase methanol:water(50:50) with addition of 0.1% of formic acid in water; transition *m*/*z* 359.95→241.96; linearity in the range of 0.2–200 ng/mL[81]dexrabeprazolesodiumRP-UPLCUV-Vis detection at λ = 284 nm;mobile phase A—phosphate buffer (pH = 7.0):acetonitrile (99:1) and mobile phase B—methanol:acetonitrile (95:5) (gradient elution)[82]ilaprazoleUPLCUV-Vis detection at λ = 305 nm;mobile phase acetonitrile:methanol:ammonium acetate buffer (0.05 M; pH = 8.5) (gradient elution); linearity in the range of 0.05–0.60 µg/mL[83]LC-ESI-MS/MSLC mobile phase 10 mmol/L ammonium formate:water-acetonitrile solution (50:50); transition *m*/*z* 367.2→184.0;linearity in the range of 0.23–2400 ng/mL[84]tenatoprazoleRP-HPLCUV-Vis detection at λ = 307 nm; mobile phase methanol:THF:acetate buffer (68:12:20); linearity in the range of 0.5–160.0 µg/mL[85]TLCstationary phase—aluminium plates with silica gel; solvent system—toluene:ethyl acetate:methanol (6 + 4 + 1), Rf = 0.34; linearity in the range of 100.0–1500.0 ng/spot[86]


## 3. PPIs’ Pharmaceutical Formulations Available on the Market

PPIs are administered by two different routes: oral or intravenous. Currently manufactured dosage forms for oral administration include enteric-coated capsules, enteric coated tablets, multiple-unit pellet system (MUPS), and suspensions with microparticulates. For intravenous administration, PPIs are available as lyophilized powders for reconstitution [53]. There are a large number of manufactured brand and generic products. The list of products available in US and EU, together with a brief characteristic, is presented in Table 3 [87,88,89].

### 3.1. Delayed-Release Tablets

Most proton pump inhibitors are available as delayed-release tablets. To protect the API from degradation, the tablet has to be coated with enteric polymers. The most commonly used polymers are methacrylate derivatives, such as a methacrylic acid copolymer with ethyl acrylate [90,91,92,93,94,95,96,97,98,99,100,101,102,103,104,105,106,107,108,109,110,111,112,113,114,115,116,117,118,119]. Another way to produce delayed-release tablets is the multiple-unit pellet system (MUPS), featuring enteric-coated pellets compressed into a tablet and coated with an immediate-release polymer. After administration, the MUPS tablet disintegrates in gastric fluids into smaller enteric-coated subunits, which are later dissolved in the small intestine. Advantages of MUPS tablets over capsules include lower manufacturing costs, smaller size, and lower risk of sticking to the esophagus during administration, due to the lack of gelatin [120,121]. Examples of MUPS tablets containing PPIs are Losec MUPS or Nexium [97,114]. A slight modification of MUPS is the modified-release orodispersible tablets (ODTs). In addition to modified-release pellets, they contain disintegrants, such as microcrystalline cellulose, which allow the tablet to disintegrate quickly in the mouth. Examples of ODT formulations include Zoton FasTab, Prevacid, and Mezzopram [122,123,124,125,126,127,128,129,130,131]. Contrary to conventional delayed-release tablets, MUPS tablets can be easily dispersed in water or other vehicle before administration, ensuring better patient compliance [124,128,131].

### 3.2. Delayed-Release Capsules

PPIs are also available in the form of hard gelatin capsules that contain pellets or granules coated with an enteric polymer [132,133,134,135,136,137,138,139,140,141,142,143,144,145,146,147]. A particular example of a formulation is Dexilant capsules, which contain double-delayed-release pellets. In this case, the first peak of the maximum plasma concentration of dexlansoprazole occurs two hours, while the second 4 to 5 h after administration [145]. Unless otherwise indicated by the manufacturer, the contents of the capsules in the form of granules or pellets can be discharged and suspended in water or applesauce to facilitate administration (Table 3).

### 3.3. Oral Suspensions

Oral suspensions are a convenient form of the drug for children, patients with swallowing problems, and those requiring enteral feeding. Formulations with PPIs usually contain enteric coated granules or pellets in sachets (Protonix) [148,149]. Some formulations additionally contain placebo pellets (Prilosec) [141,150,151] or have powdered form (Zegerid) [152,153,154]. Oral suspensions of proton pump inhibitors are prepared ex tempore. Depending on the manufacturer’s recommendations, they can be administered with water or with applesauce or fruit juice [148,150,152]. A study by Bladh et al. [155] assessed the stability of esomeprazole magnesium manufactured in the form of sachets containing pellets intended to form oral suspension in water, apple sauce, and apple or orange juice. The suspension was found to remain stable after preparation in the pH range of 3.4–5.0 for up to 60 min. The time needed to disperse pellets in water was 2 min, while in juices it was ca. 15 min. After the pellets were dispersed in apple or orange juice, their stability was above 98%. During delivery of the suspension through the feeding tube, more than 96% of the drug substance was delivered. The formulation was bioequivalent to tablets and capsules containing the same drug substance [155]. Johnson et al. [156] also tested the stability of enteral pellets containing esomeprazole in 100 mL of water, milk (1.5% fat), cultured milk, yogurt and apple or orange juices. After 30 min of incubation, dissolution studies were performed in hydrochloric acid and phosphate buffer. The loss of esomeprazole was less than 2% in all the media tested except milk (with 1.5% fat).

In hospital pharmacies, suspensions containing PPIs are also prepared by dissolving the formulations in a solution of 8.4% sodium bicarbonate, or alternatively in ready-to-use compounding media, e.g., Alka’s SyrSpend SF. Polonini et al. [157] demonstrated that such medium can be used to prepare suspensions containing omeprazole, pantoprazole, esomeprazole, and lansoprazole. It is a dry compounding base in the form of a powder ready for reconstitution, containing modified food starch, calcium carbonate, and sucralose. The vehicle is characterized by its taste-masking effect, lack of preservatives and sodium addition, as well as high microbiological stability [157].

In addition, oral liquids can be prepared by crushing tablets and suspending them in vehicles mentioned above. This method was described by Dentinger et al. [158] in a study on the stability of pantoprazole in oral liquid suspensions. The researchers found that the oral liquid prepared extemporaneously from pantoprazole tablets in an 8.4% sodium bicarbonate solution was stable in dark glass bottles for 62 days under refrigeration conditions [158]. Melkoumov et al. [159] investigated the stability of lansoprazole microgranules (Prevacid FasTab) after suspending them in the Ora-Blend^®^ vehicle. Ora-Blend^®^ is a flavored colloidal medium with slightly acidic pH. It was found that the extemporaneous formulation with lansoprazole remained stable for 3 days at 4.5–5.5 °C [159]. Ferron et al. [160] reported the results of an open-label, randomized, two-period crossover study comparing the bioavailability of pantoprazole administered as a suspension in 8.4% sodium bicarbonate solution and as delayed-release tablets. It was observed that the bioavailability of pantoprazole from suspension was lower, although both formulations reached similar maximum concentrations [160].

### 3.4. Powders for Injections or Infusions

Due to the poor solubility of PPIs, their salts are used to formulate intravenous dosage forms. Examples include sodium salts of omeprazole, esomeprazole, and pantoprazole. They are marketed as lyophilized powders for infusions or injections [161,162,163,164,165,166,167,168]. In the manufacturing process, the alkaline aqueous solution of PPI is adjusted to pH 11 with sodium hydroxide. Subsequently, the solution is filtered and then dispensed into 5 or 10 mL vials and freeze-dried under aseptic conditions. The recommended solvents for reconstitution are 0.9% sodium chloride or a 5% dextrose solution. The least commonly used is the lactated Ringer’s solution (e.g., Nexium I.V.). The pH of the reconstituted solution for infusion is usually around 10 [161,162,163,164,165,166,167,168]. Omeprazole sodium is also available as a combination product for injection. The product contains a vial with lyophilized omeprazole powder and a separate solvent ampoule. The solvent is composed of water, macrogol 400 and citric acid. The pH of the solution after reconstitution is around 8.6 [161]. Reconstituted solutions are stable for a short period of time, usually 6 to 12 h at room temperature, and must not be stored. Pantoprazole in injection solutions has been reported to be three times more stable than omeprazole [169].

The stability of esomeprazole sodium at concentrations of 0.4 and 0.8 mg/mL was investigated in three injection solutions: 0.9% sodium chloride, 5% dextrose and lactated Ringer’s solutions. It was stable in all of them for 48 h at room temperature and 120 h under refrigeration [170].

In the case of infusion solutions, Carpenter et al. [171] found that omeprazole and pantoprazole can be stored for up to 48 h at room temperature without significant loss of drug substance (less than 6%). The powder for infusion reconstituted with a 5% dextrose solution was less stable than with a 0.9% sodium chloride solution [171]. Johnson [172] proved that a solution of pantoprazole sodium in 0.9% sodium chloride stored in polypropylene syringes remained stable for 96 h, both at room temperature and in the refrigerator. After three days of storage at room temperature, the solution turned a slight yellow-orange color, but HPLC analysis showed no unacceptable changes in drug substance content [172]. However, due to the high risk of potential interaction, reconstituted solutions of PPIs cannot be administered with parenteral nutrition or mixed with other infusion solutions [161,162,163,164,165,166,167,168].
pharmaceutics-14-02043-t003_Table 3Table 3Examples of proton pump inhibitors currently marketed in the US and EEA (European Economic Area)/UK [87,88,89].PPIDosage [mg] *Drug FormBrand Name (e.g.)Additional CommentsReferencesOmeprazole10, 20, 40DR tablets Omeprazole Dexcel PharmaEC polymer: hypromellose acetate succinate,DR tablets approved by FDA contain 20 mg of omeprazole (OTC only)[90,91,92]20Orally disintegrating DR tabletsOmeprazole DR Orally Disintegrating tablets Dexcel PharmaContains aminomethacrylate copolymer and hypromellose phthalate[122,123]10, 20, 40Capsules with DR pelletsLosecHard gelatine capsules containing EC pellets (methacrylic acid—ethyl acrylate copolymer (1:1) dispersion 30%),Content of the capsule can be mixed with fruit juice, applesauce, or in non-carbonated water before swallowing[132,133,134]Omeprazole SandozHard gelatine capsules containing EC pellets (methacrylic acid copolymer dispersion type C/hypromellose phthalate (40 mg capsules)),Content of the capsule can be mixed with tablespoon of applesauce before swallowing[135]20, 40Oral Suspension, DRZegeridCombination product (omeprazole + 1.68 g sodium bicarbonate),Powder should be suspended in 5–10 mL of water,Available in capsules (20 mg, 40 mg of omeprazole),Can be administered via NG/OG tubes (with 20 mL of water)[152]2 mg/mL4 mg/mLOmeprazole, Powder for Oral SuspensionContains potassium hydrogen carbonate and sodium hydrogen carbonate,Powder should be suspended with 64 mL water to obtain 180 mg/360 mg of omeprazole per bottle respectivelyAfter reconstitution suspension can be stored for 28 days at 2 °C–8 °CCan be administered via NG/PEG tubes[153,154]Omeprazole magnesium20DR tabletsPrilosec OTC20.6 mg of omeprazole magnesium is equivalent to 20 mg of omeprazoleEC polymer: Methacrylic Acid–Ethyl Acrylate Copolymer (1:1) Dispersion 30%[93,94]Losec Control[95]10, 20, 40Losec MUPS10.3 mg/20.6 mg/41.3 mg of omeprazole magnesium is equivalent to 10 mg/20 mg/40 mg of omeprazole respectivelyTablets containing EC micropellets; EC polymer: Methacrylic Acid–Ethyl Acrylate Copolymer (1:1) Dispersion 30%,Tablets can be broken and dispersed in a tablespoon of non-carbonated water, fruit juices, or applesauce[96,97,98,99]Orally disintegrating DR tabletsMezzopramOrodispersible tablets composed of GR pellets,10.3 mg/20.6 mg/41.3 mg of omeprazole magnesium is equivalent to 10 mg/20 mg/40 mg of omeprazole respectively,EC polymer: Methacrylic acid-ethyl acrylate copolymer (1:1)Can be dispersed in tablespoon of non-carbonated water, applesauce or fruit juices, or administered via PEG[124,125,126]20Capsules with DR pelletsOmeprazole Magnesium20.6 mg of omeprazole magnesium is equivalent to 20 mg of omeprazole,EC polymer: methacrylic acid copolymer dispersion and methacrylic acid copolymer Type B[136]Omeprazole Magnesium DR mini-capsulesDR mini-capsules approved by FDA in May 2022[147]2, 5, 10Oral suspension, DRPrilosec Granules should be suspended in 5 mL/15 mL of water respectively,Contains EC (brownish) and inactive (yellow) granules,EC polymer: methacrylic acid copolymer C,Can be administered via NG/gastric tube[150]Omeprazole sodium40Powder for solution for infusionOmeprazole 40 mg Powder for Solution for InfusionContains sodium hydroxide for pH stabilization,42.6 mg of omeprazole sodium is equivalent to 40 mg of omeprazole,After reconstitution 1 mL contains 0.4 mg omeprazole[167]Pantoprazole sodium sesquihydrate20, 40DR tabletsControloc (Pantoprazole sodium Takeda)Controloc Control 20Contains 20 mg/40 mg of pantoprazole as pantoprazole sodium,EC polymer: Methacrylic acid-ethyl acrylate copolymer (1:1),Controloc Control is an OTC drug[100,101,102]Protonix[148]40Oral suspension DRProtonixPantoprazole SUN PharmaGranules should be sprinkled in one teaspoon applesauce or apple juice only,EC polymer: methacrylic acid copolymer,Can be administered via NG/gastric tube[148,149]40Powder for solution for injectionProtium I.V.Pantoprazole 40 mg ZentivaOne vial contains 40 mg of pantoprazole in form of pantoprazole sodium,Contains sodium hydroxide for pH stabilization,Can be reconstituted with 0.9% NaCl or 5% glucose solution[165,166]Protonix I.V.Freeze-dried powder in single-dose vial for reconstitution,Contains sodium hydroxide for pH stabilization[164]Lansoprazole15, 30Orally disintegrating DR tabletsZoton FasTabLansoprazole MylanOrodispersible tablets composed of GR microgranules,EC polymer: methacrylic acid—ethyl acrylate copolymer (1:1) dispersion 30 percent,Can be administered with a sip of water or dispersed in a small amount of water or administered via NG tube/oral syringe[127,128,129,130]PrevacidOrodispersible tablets composed of GR microgranules,EC polymer: methacrylic acid,Can be administered after dispersion in 4 mL/10 mL of water respectively or via NG tube/oral syringe[131]Capsules with DR pelletsLansoprazole AccordEC spherical microgranules/pellets in a hard gelatine capsule,EC polymer: Methacrylic Acid-Ethyl Acrylate Copolymer, 1:1, Dispersion 30%,Microgranules can be mixed with a small amount of water, apple/tomato juice, or sprinkled onto a small amount of soft food,Can be administered via NG tube[137,138]Lansoprazole Capsules SandozDR pellets in hard gelatine capsule,EC polymer: methacrylic acid copolymer dispersion,Pellets can be sprinkled on the one teaspoon of applesauce, pudding, cottage cheese, yogurt, or strained pears or sprinkled in 60 mL of apple, orange, or tomato juice,Can be administered via NG tube[146]Rabeprazole sodium 10, 20DR tabletsParietRabeprazole Accord10 mg/20 mg rabeprazole sodium is equivalent to 9.42 mg/18.85 mg rabeprazole,EC polymer: Methacrylic acid-ethyl acrylate copolymer with undercoating composed of ethylcellulose and magnesium oxide[103,104,105,106]20AciphexRabeprazole sodium AurobindoEC polymer: hypromellose phthalate,Can be administered with or without food[107,108]Esomeprazole magnesium20, 40DR tabletsNexiumEsomeprazole AccordContains GR pellets,EC polymer: methacrylic acid ethyl acrylate copolymer (1:1) dispersion 30 per cent,Nexium contains esomeprazole as esomeprazole magnesium trihydrate (22.3 mg, 44.5 mg respectively),Esomeprazole Accord contains esomeprazole as esomeprazole magnesium dihydrate 21.75 mg, 43.5 mg respectively),Tablets can be dispersed in half a glass of non-carbonated water,Can be administered via gastric tube[109,110,111,114]20Nexium 24HEsomperazole Dr Reddy’sEC polymer: Methacrylic acid-ethyl acrylate copolymer (1:1),Contains esomeprazole as esomeprazole magnesium trihydrate[112,113]20, 40Capsules with DR pelletsVentraEC granules in hard gelatine capsule,Contains esomeprazole as esomeprazole magnesium dihydrate (21.75 mg, 43.5 mg respectively),[139,140]NexiumEC granules in hard gelatine capsule,EC polymer: methacrylic acid copolymer type C,Contains esomeprazole as esomeprazole magnesium trihydrate (22.3 mg, 44.5 mg respectively),Can be taken sprinkled onto one tablespoon of apple juice,Can be administered via NG/gastric tube[141]2,5, 5, 10, 20, 40Oral suspension, DRNexiumContains esomeprazole as esomeprazole magnesium trihydrate (2.8 mg, 5.6 mg, 11.1 mg, 22.3 mg, 44.5 mg respectively),Brownish granules contain API; yellow granules are inactive,EC polymer: Methacrylic acid–ethyl acrylate copolymer (1:1) dispersion 30%,The dosage of 2.5 and 5 mg/sachet should be dispersed in 5 mL of water; 10–40 mg/sachet in 15 mL of water and left for 2–3 min to thicken,Can be administered via NG/gastric tube[141,151]Esomeprazole sodium20, 40GR tabletsEsomeprazol CinfaContains GR pellets,EC polymer: Methacrylic acid –ethyl acrylate copolymer (1:1) dispersion 30%,Can be administered via gastric tube[115,116]20, 40GR capsulesEsomperazol CinfaContains EC spherical granules,EC polymer: copolymer of methacrylic acid—ethyl acrylate and triethyl citrate,Can be administered via gastric tube[142,143]40Powder for solution for infusion/injectionNexium IV40 mg esomeprazole is equivalent to 42.5 mg esomeprazole sodium,Lyophilized powder in a single-dose vial for reconstitution[162,163]Dexlansoprazole30, 60Capsules with DR pelletsDexilantDR granules in hard gelatine capsule (pH-independent dissolution),Functional polymers: methacrylic acid copolymers,Dual release formulation- first plasma concentration peak occurs within 1–2 h and second in 4–5 h after administration,Can be sprinkled on one tablespoon of applesauce or suspended in 20 mL of water for administration via NG tube/oral syringe,Can be administered with or without food[144,145]Ilaprazole5, 10, 20DR tabletsNoltecNorutec
[117,118,119]GR—gastro-resistant; EC—enteric-coated/-ing; DR—delayed release; NG/OG/PEG tubes—nasogastric/orogastric/percutaneous endoscopic gastrostomy (PEG) tubes for enteral nutrition; * calculated on basic form excluding rabeprazole sodium.


### 3.5. Fixed-Dose Combinations

Some PPIs are also available as fixed-dose combination products (Table 4). These can be divided into groups depending on the indication for use. The first group includes PPIs combined with non-steroidal anti-inflammatory drugs (NSAIDs) such as diclofenac or naproxen. These products are marketed as delayed-release tablets (Vimovo) or capsules filled with pellets (Diclopram). Indications for use include an increased risk of gastric and/or duodenal ulcers after the use of NSAIDs. In this combination, PPI is used at the standard dose of 20 mg [173,174]. The second group consists of products marketed as kits for the treatment of Helicobacter pylori infection. The kit is a combination pack containing PPI, antibiotics, and/or antimicrobials packed in blisters sufficient for one-day dosing. The drug substances and dosages in the package were selected according to the standard recommendations for the eradication of H. pylori infections. For instance, Panclamox 40/500/1000 consists of 14 sets of three tablets with pantoprazole 40 mg, clarithromycin 500 mg and amoxicillin 1000 mg. One blister contains tablets of each formulation, sufficient for one-day dosing [175]. The third category includes products used for the treatment of GERD, which contain PPI and sodium bicarbonate, such as Zegerid. Sodium bicarbonate increases the pH in the stomach and thus protects omeprazole from degradation. Therefore, the formulation does not contain enteric polymers. Zegerid is available in the form of capsules or powder for oral suspension. The formulations differ in the overall sodium content and, due to this, the products are not interchangeable [152]. Examples of fixed-dose combination products are presented in Table 4.

### 3.6. Administration of PPIs via a Feeding Tube

Proton pump inhibitors can be administered to adult and pediatric patients who require enteral nutrition via a feeding tube. However, factors such as the risk of clogging the tube or adhering the drug to the walls of the tube or the possibility of drug degradation should be carefully considered before administration [141,180]. The most convenient dosage forms for application via feeding tube are those composed of pellets or granules that can be easily dispersed in water or other vehicles. These include tablets or capsules containing delayed-release pellets, as well as granules for oral suspensions (Table 3). PPIs should be administered via feeding tube only in accordance with the manufacturer’s recommendations. An example of a drug that can be administered via the nasogastric or gastric tube is Nexium, a delayed-release oral suspension. To prepare the dose, a catheter-tipped syringe should be filled with 5 mL of water. The content of the packet with 5 mg of esomeprazole should be added to the syringe, shaken, and left for 2–3 min to thicken. If the granules have dissolved or disintegrated, the dose should be discarded. The mixture should be administered in 30 min through a tube of size 6 Fr or larger. Finally, the tube should be flushed from any remaining contents of the drug. It is not recommended to mix the dispersion of PPI before administration with other drugs and solutions [130].

### 3.7. Pediatric Population

According to data available in the UptoDate database [181] and the summaries of product characteristics (SmPC) collected in Table 3, proton pump inhibitors are used to treat GERD and erosive esophagitis in pediatric patients. Omeprazole, esomeprazole, and lansoprazole are used in the therapy of patients under 1 year of age. In children older than 5 years, pantoprazole can be used, while dexlansoprazole and rabeprazole are indicated for patients older than 12 years. For pediatric patients, the drug dose is calculated per kilogram of body weight (usually for infants) or expressed as a range of weight to the drug dose. Generally, it is assumed that the dose of the drug given to a child should not exceed the maximum dose for an adult [181].

Among PPIs, only esomeprazole is registered in the form of powder for solution for infusion dedicated to pediatric patients. Other forms of drugs used for pediatric patients include capsules, orally disintegrating tablets, film-coated tablets, MUPS tablets, and granules for oral suspension. To simplify administration, the contents of capsules or a sachet containing enteric-coated granules can usually be sprinkled on soft food or, like ODT, suspended in water or fruit juice. An oral syringe can be used for easier administration of the drug in an aqueous dispersion. Enteric coated tablets used in pediatrics usually have a small diameter and should not be crushed or chewed due to the protective layer. PPIs can also be administered to children via a nasogastric or enteral tube. Ponrouch et al. [182] tested the effect of parameters such as tube opening and length, volume of the PPI diluent, volume of the tube flush solvent, and the form of the drug on the efficiency of delivery of PPIs to children via nasogastric tube. Among the solid oral dosage forms tested, only ODTs appeared to be administered without any loss via the 8 Fr tube [182].

Another product for pediatric administration is a FIRST kit for the prescription compounding of a flavored oral suspension of omeprazole or lansoprazole produced by Azurity Phamaceuticals. The FIRST kit consists of a bottle containing pre-weighted powder of the drug substance, a bottle with the ready-to-use suspending solution, an adapter cap, and an oral syringe. The concentration of omeprazole and lansoprazole after preparation is 2 mg/mL and 3 mg/mL, respectively. The solution contains sodium bicarbonate, strawberry flavor, and stabilizers. The reconstituted drug can be stored for at least 30 days in the refrigerator. Available bottle sizes are 90, 150 and 300 mL. The kit is intended for compounding in pharmacies only. The advantages of the product are ease of use, pre-measured components, and short preparation time [183,184,185].

According to the EMA 2015 Inventory of pediatric therapeutic needs, there is a need to provide more data on pharmacokinetics, safety and efficacy of PPIs in pediatric treatment, particularly in parenteral and gastric use.

### 3.8. Storage and Packaging

Davidson et al. [186] published a survey on the stability of omeprazole products from 13 countries. During the study, 31 capsule products and 3 types of tablets originated from Sweden, Korea, China, Turkey, Spain, India, Mexico, Brazil, Argentina, Greece, Portugal, Thailand, and Chile were tested. The products have been subjected to stability tests under accelerated storage conditions (40 °C, 75% RH) for 6 months. Among the investigated drugs, 27 failed to meet the assumed stability criteria. The defined standards included an almost unchanged appearance, dissolution profile and API content, plus less than 1% of impurities after 6 months of storage. Darkening of the white/pale-brown samples indicated the degradation of omeprazole during storage. Only six products (including the original drug, Losec Capsules, Astra) met the study criteria. The results of the study raised concerns about the quality and safety of the investigated products [186]. Although the stability of the drugs is usually predicted based on long-term tests, poor results in short-term studies may or may not indicate long-term storage problems. However, to draw the correct conclusions from the study, more detailed information is needed on the packaging, recommended storage temperature, and coating polymers. Stroyer et al. [46], confirmed a higher risk of interaction between PPIs and enteric polymers when stored under accelerated conditions.

According to the summaries of product characteristics for the drugs presented in Table 3, products should not be stored above 25–30 °C. The expiration date for oral formulations of proton pump inhibitors is 3 years. Powders for solutions for infusion can be stored for 2 years. However, the shelf-life of suspensions and solutions after reconstitution shortens to 30 min, or 6–12 h, respectively (Table 3). The packaging of PPIs should guarantee barrier properties against moisture. For instance, Nexium sachets, which contain esomeprazole granules for oral suspension, consist of three-layer material, protecting the drug against moisture. The laminate is composed of PET (polyethylene terephthalate), aluminum, and LDPE (low-density polyethylene) [151]. Tablets and capsules are packed in bottles or blisters. Blister strips are formed of aluminum in the process of cold-forming [103,104,127,128]. The advantage of Alu/Alu blisters is nearly complete protection against moisture, light, and oxygen. Another encountered blister composition is polyamide/aluminum/PVC [95]. Bottles are usually made of high-density polyethylene (HDPE) or PET [133,153,154].

## 4. Development of New Formulations with PPIs

Proton pump inhibitors have been marketed worldwide for more than 30 years [9,11]. At this time, many pharmaceutical solutions have been proposed to improve their acceptability, stability, safety, and efficacy (Table 5). The first proton pump inhibitor developed and marketed was an omeprazole. Although there are a dozen other PPIs, omeprazol still remains the most popular in scientific work conducted on the different aspects of antisecretory drugs. The search carried out in the Web of Science Core Collection database revealed more than 15,000 items with the term ‘omeprazole’, which is even more than the total number of items for all other PPIs (Figure 3).

The most popular route of administration of PPIs is oral, which is the most common for all medicinal products. Formulations with PPIs include numerous dosage forms, starting from simple enteric-coated tablets or pellets encapsulated in hard gelatine capsules, through the other novel form of tablets, and ending with many different forms of micro- or nanoparticulates. There have also been some approaches to the administration of PPIs through alternative routes of administration, such as transdermal or rectal.

### 4.1. Nanoparticles

Nanoparticles have received a lot of attention in recent years due to the fact that they are biodegradable and biocompatible. They are finding application in a targeted therapy as well as providing modified release of the therapeutic substance. In this group of formulations, the most common are nanoparticles and nanocapsules, for which various technological solutions have been developed.

Bendas et al. [187] developed a method to produce gastro-resistant nanocapsules loaded with omeprazole based on the hydroxypropylmethylcellulose phthalate (HPMCP) or polyvinyl acetate phthalate (PVAP) in different ratios (1:2.5, 1:5, and 1:10). To obtain the nanocapsules, an organic phase containing omeprazole, lecithin, and enteric polymer dissolved in a mixture of acetone and ethanol was suspended in an aqueous solution of poloxamer (surfactant) and sodium bicarbonate. Miglyol^®^ 812 (oil) was added to the mixture, the suspension was evaporated at 45 °C for 15 min and then lyophilized. The nanocapsules had a spherical shape, a smooth surface, and a diameter in the range of 200–500 nm. The encapsulation efficiency between 31 and 65% increased along with the amount of polymer used. Formulation with the best gastro-retentive properties, i.e., containing polymer in 10:1 ratio to drug substance, was administered to the male Winstar rats, which proved the anti-ulcer activity of the obtained nanocapsules [187].

In another study, nanoparticles were produced using Eudragit RS100 as a matrix-forming polymer. The diameter of the obtained particles ranged from 199 to 370 nm, and increased with the higher amount of polymer. On the other hand, the higher amount of polymer caused a lower drug loading in the particles. The nanoparticles released lansoprazole for 24 h by a diffusion mechanism combined with swelling of the polymer chains [188].

Promising studies were conducted by Nasef et al. [189] and Rezazadeh et al. [190]. The authors attempted to obtain pH-dependent nanoparticles from enteric polymers. These formulations were developed to protect drug substances from premature release in the stomach, and thus their degradation. The authors of the first study produced Eudragit L100-55 and chitosan nanoparticles using a complex coacervation method. The diameter of the obtained nanoparticles was approximately 800 nm, and they have shown good gastric resistance while releasing omeprazole for up to 12 h in intestinal fluids. The pharmacological efficacy of the nanoformulation was proven during in vivo studies in rats [190]. Nasef et al. [189] used Eudragit S100 and hydroxypropylmethylcellulose phthalate (HP-55) to produce pH-dependent nanoparticles with pantoprazole. Depending on the formulation, the size of the particles was 300 or 640 nm. For both formulations, the nanoparticles were acid-resistant, but those made with HP-55 had a slower release of pantoprazole [189]. Another method of producing pantoprazole nanoparticles was proposed by Sheikh et al. [191]. The researchers used a solid lipid injection method to produce 12 formulations with different contents of ethyl cellulose, chitosan and HPMC, respectively. To obtain pantoprazole particles, matrix-forming polymer and polyvinyl alcohol were dissolved in ethanol heated to 70 °C. Then it was added drop by drop to the warm phosphate buffer solution with dichloromethane. Nanoparticles precipitated after sonification of the mixture. Nanoparticles containing 30% chitosan had the highest entrapment efficiency and drug content. Pantoprazol was released for up to 12 h from such formulations [191].

Other formulations based on the production of nanoparticles include nanosponges. These are porous nanostructures formed by free cavities filled with a therapeutic substance. Due to their spongy structure, they are mainly used for topical and controlled delivery of the drug. In the case of oral formulations, they may be mixed with other excipients to form sustained-release tablets or capsules. Penjuri et al. [192] used an emulsion solvent diffusion technique to prepare nanosponges with lansoprazole and ethylcellulose characterized by extended release of up to 12 h [192].

Nanoformulations developed with PPIs also included nanofibers. Karthikeyan et al. [193] prepared nanofibers containing a combination of aceclofenac with pantoprazole. A major advantage of nanofibers over other drug formulations is the high release rate of the therapeutic substance and the high efficiency of the manufacturing process. To obtain the nanofibers, a 20% *w*/*v* ethanolic solution of zein and a 10% *w*/*v* methanolic solution of Eudragit S100 were prepared in which aceclofenac and pantoprazole were dissolved, respectively. After these two solutions were mixed, electrospinning was initiated. A voltage of 25 kV was applied to a metal needle with an inner diameter of 0.5 mm and the solution flow rate was set at 1 mL/h. The process was carried out at 24 ± 1 °C. It was shown that the drug substances have an amorphous form and were uniformly dispersed in the polymer matrix. Nanofibers with a smooth surface had a diameter of 50–200 nm. The encapsulation efficiency was ca. 35% and 66% for aceclofenac and pantoprazole, respectively. Dissolution studies have shown that 6% pantoprazole and 25% aceclofenac were released to a hydrochloric acid solution. The release of both therapeutic substances was prolonged to 8 h at the pH = 7.4. Furthermore, animal studies showed that the gastric mucosa of rats was intact after administration of the nanofibers produced [193].

Proton pump inhibitors were also formulated in the form of nanoemulsion systems. Ahuja et al. [194] developed an immediate release formulation of lansoprazole-containing niosomes, while Mohanty et al. [195] prepared proniosomes with omeprazole. Other studies included the preparation of nanosuspensions for which lansoprazole complexes with B-cyclodextrins were used [196] and the application of bioactive solid self-nanoemulsifying drug delivery systems for the delivery of lansoprazole and curcumin [197].

### 4.2. Microparticles

Microparticles are made of biopolymers in which the therapeutic substance is encapsulated or uniformly dispersed. Their advantages include the ability to improve the stability and bioavailability of the drug substance, modification and control of its release site and rate. The most common methods of microparticle production are the solvent evaporation or emulsification method [51,52,198,199,200,201,202,203,204,205,206,207,208,209,210,211,212,213,214] and spray-drying [199,200,215,216,217,218,219,220]. The advantage of the latter over the other methods is the ease of process scale-up [215].

Raffin et al. [209] developed double-walled microparticles using the two-step emulsification method. Pantoprazole was microencapsulated using biodegradable polymer poly(ε-caprolactone) and subsequently coated with Eudragit S100 to provide the colonic delivery of the drug substance. The microparticles were characterized by only partial protection of pantoprazole in an acidic medium (approximately 30%). However, after compression into tablets, an increase in the stability of PPIs and a controlled release of pantoprazole from the microparticles was observed [209]. Boddupalli et al. [221] prepared gastro-resistant microparticles containing omeprazole and piperine using a solvent evaporation technique. The excipients used for the formation of microspheres were ethylcellulose, HPMC, and calcium carbonate. The drug substances and excipients were dissolved in acetone and then transferred to liquid paraffin containing the emulsifier Span 80 (3%). The emulsion was stirred until the solvent evaporated, and then the microspheres obtained were purified with petroleum ether. Microspheres containing a combination of omeprazole with piperine exhibited much better pharmacokinetic performance in vivo (in rabbits) than those with omeprazole alone [221].

Among the most commonly used polymers for the manufacturing of microparticles containing proton pump inhibitors are methacrylate derivatives such as Eudragit S and L [200,202,209,211,215,216,217,219,222], as well as Eudragit RS/RL [199,205,206,214,220]. Although Eudragit S and L are used to make delayed-release forms of the drug, they can also be used as polymers to formulate sustained-release microparticles [200,202,209,211,215,216,217,219,222]. In addition, cellulose derivatives such as ethyl cellulose, HPMC or hydroxypropylmethyl cellulose phthalate are also utilized [198,203,204,207,208,213,223]. The work of Singh et al. [212] provides an example of the development of microparticles containing lansoprazole based on chitosan and γ-poly-(glutamic acid). An ionic-crosslinking reaction between chitosan and sodium tripolyphosphate (STPP) followed by freeze-drying was used for the microparticles preparation. They did not show acid-resistance, therefore they were finally placed in an enteric-coated capsule [212]. Qamsari et al. [224] attempted to encapsulate omeprazole in S-layer proteins produced by the bacterial strain Lactobacillus acidophilus ATCC 4356. These proteins have the self-assembling ability to interact with each other through non-covalent bonds to form a monomeric layer 5–20 nm thick. In addition, when they are exposed to acids, they have the ability to reconstitute their structure. It was observed that coating of the omeprazole particles with lactobacillus S-layer proteins reduced its degradation in acetate buffer at pH = 5 [224].

Another approach to increase the stability and solubility of omeprazole was to encapsulate it in complexes with cyclodextrins or ion-exchange resins. Cyclodextrins are cyclic oligosaccharides that have a wide range of pharmaceutical applications. β-cyclodextrins consist of 7 α-d-glucopyranose molecules linked by an α-1,4-glycosidic bond, giving them a torus-like shape. The inner part of cyclodextrins is hydrophobic, and the outer part is hydrophilic. Therefore, they are used to increase the solubility and stability of medicinal substances, taste masking, etc. [225]. Loftsson et al. [226] studied the effect of complex formation with HP-β-CD on the water solubility and stability of 13 drugs, including omeprazole. The degradation of omeprazol complexed with cyclodextrin was 1.1- to 2.5-fold slower than in uncomplexed form [226].

To improve the stability of lansoprazole, studies were conducted with the aim of incorporating it into cyclodextrin metal-organic frameworks (CD-MOFs). High uniform saturation of the complexes with the therapeutic substance was successfully achieved. The microparticles had a diameter of approximately 5 µm and retained thermal stability [227]. Ruiz et al. [228] performed studies on omeprazole complexes with latex particles that improve stability in acidic conditions. For this purpose, Aquateric^®^, an aqueous dispersion of cellulose acetophthalate latex was used. The reaction of adsorption of the drug substance on the latex carrier occurs under acidic conditions, where the latex remains stable. However, in solutions with a pH above 6.5, the electrostatic charge on the surface of latex particles changes, causing the resulting complexes to rapidly disintegrate and release the therapeutic substance [228].

Patents EP 0 998 308 B1, WO 2004/060357 and WO 00/40224 report the formation of PPI complexes with ion exchange resin to increase its stability under the influence of acidic moieties of coating polymers [229,230,231]. Ion-exchange resins are insoluble polymers that have the ability to exchange ions in aqueous media because of the presence of cationic or anionic moieties in the polymer structure. As cationic resins require an acidic environment for ion exchange, anionic resins, such as cholestyramine, are used for acid-labile PPIs [229,232]. For example, the patent EP 0 998 308 B1 [229] claims a method of production of enteral formulations for benzimidazole derivatives based on complex formation with an anion exchange resin. The inventors declared that the complexes between the benzimidazole derivatives and cholestyramine provide adequate stability to the substance during coating with low acidic enteric polymers and during storage. As a result, it is possible to use a hydroxypropylmethylcellulose phthalate coating (HP-50) without the need for an additional water-soluble protective layer application [229].

In order to improve the solubility and release rate of omeprazole from microparticles, El-Badry et al. [233] conducted studies on solid dispersions of omeprazole. For this purpose, they used Kollidon IR and β-cyclodextrins. The microparticles containing each substance were obtained by spray drying or lyophilization, and placed in enteric-coated capsules. The release profiles of the produced formulations were compared with those of omeprazole powder. The omeprazole contained in the manufactured microparticles was shown to be converted from a crystalline to an amorphous form, which significantly improved its solubility. Optimized formulations based on Kollidon IR with cyclodextrins showed acid resistance and rapid release in phosphate buffer medium within 40 min, thus fulfilling the criteria for delayed release [233]. The conversion of omeprazole into an amorphous form by forming complexes with cyclodextrins was also observed in other studies by the same author [234].

Geng et al. [29] also used B-cyclodextrins as a stability enhancer for omeprazole. The aim of the study was to develop a capsule containing 20 mg of omeprazole and antacids with a reduced powder content and therefore a smaller size. Omeprazole-β-CD complexes were produced by dissolving cyclodextrins in a sodium hydroxide solution at pH = 11. Then, omeprazole was added to the solution and stirred at an elevated temperature. The highest encapsulation efficiency (47%) was obtained using a ratio of 1:2 API to β-CD, a temperature of 60 °C, and 3 h of stirring. The hydroxypropylcellulose capsules were then filled with the resulting complexes and antacid excipients, finally containing 250 mg of sodium bicarbonate and 400 mg of magnesium oxide, while a comparable commercial capsule contained 1100 mg of sodium bicarbonate. Release studies under simulated gastric acid conditions showed that the release profile of omeprazole in both the formulation and the commercial preparation differed significantly. The commercial preparation appeared to gradually release about 60% of omeprazole. After reaching a maximum, its content dropped to zero after 45 min of testing. On the contrary, the manufactured formulation showed rapid release of the therapeutic substance of up to 90%. The high peak concentration persisted for an extended period, after which the amount of omeprazole was dropped to zero. This observation is related to the composition of the capsule content. The antacids used in the capsule allowed the simulated gastric juice to be maintained above pH = 7 for 40 min. In vivo studies in rabbits proved that the maximum plasma concentration of omeprazole was higher with the obtained capsules than with the commercial preparation [29].

A separate subcategory includes mucoadhesive and floating microparticles. They are designed to provide prolonged or controlled release of PPIs while the drug floats on the surface of gastric juice [202,235]. Due to this, floating microparticles are also referred to as microballoons [211,214]. Extended-release matrix polymers, such as cellulose derivatives, are used to produce gastroretentive microparticles. For example, Muthusamy et al. [213] developed floating lansoprazole micropellets. They were obtained by the emulsification method, using three carrier polymers: chitosan, methylcellulose, and hydroxypropylmethylcellulose. The drug to polymer ratio applied was 1:1, 1:2, and 1:3. Lansoprazole and polymer were dissolved in the mixture of ethanol and dichloromethane, then dropped into the polyvinyl alcohol solution and stirred for one hour. The size of the obtained micropellets was in the range of 327–431 µm. The floating properties of the microparticles were evaluated under simulated gastric conditions for 12 h and all of the formulations were characterized by great buoyancy. The highest drug entrapment ratio (93%) was attributed to the formulation of chitosan with a drug to polymer ratio of 1:3. Micropellets containing chitosan in a 1:1 drug to polymer ratio showed 72% drug release in phosphate buffer at pH = 6.8, which was the highest value among the investigated formulations [213]. However, the most commonly used polymers in this category are methacrylate derivatives, such as Eudragit L [211], Eudragit S [202,214], and Eudragit RS [206]. Masareddy et al. [235] developed flotation microbeads containing rabeprazole. They were prepared from sodium alginate using the ionotropic gelation method. Calcium chloride and alternatively barium chloride were used as crosslinking agents, while sodium bicarbonate was used as the gas-forming excipient. The produced microparticles released the therapeutic substance for up to 10 h. Calcium chloride was found to have better crosslinking properties [235]. On the contrary, Sheikh et al. [208] developed mucoadhesive microparticles with clarithromycin and omeprazole for the treatment of Helicobacter pylorii. The microparticles for each substance separately were produced by emulsification solvent evaporation. Carbopol 971p and HPMC K4M or K100M were used as mucoadhesive polymers. The optimized formulations with the best performance contained Carbopol and HPMC K100M in a ratio of 1:1. The microparticles were characterized by a size of 257–370 µm, smooth surface, and a spherical shape. When placed in capsules, they provided the release of clarithromycin and omeprazole for up to 8 h [208].

### 4.3. Minitablets

Minitablets have been developed as a multi-compartment drug dosage form. They are tablets with a diameter of 1 to 3 mm designed primarily for pediatric patients. Filho et al. [236] and Szczepanska et al. [237] attempted to develop minitablets containing PPIs.

Filho et al. [236] developed gastro-resistant minitablets containing omeprazole. Two series of core minitablets were obtained. The main excipient in formulation A was microcrystalline cellulose, whereas in formulation B the main excipient was spray-dried lactose. Minitablets were coated in a fluidized bed with an insulating layer composed of 4% hydroxypropylmethylcellulose solution until 3% of the weight gain of the tablet was achieved. Subsequently, the minitablets were coated with enteric-release polymer Eudragit S100 D55 to obtain the 8, 10, and 12% weight gain of the tablet, respectively. The minitablets were packed in hard gelatin capsules, in the amount equal to 20 mg of omeprazole per capsule. The drug release profiles of the developed formulations were compared with those of the reference product. Minitablets coated with Eudragit to achieve 8% weight gain have not met the gastro-resistance criteria. Other series were sufficiently coated with enteric polymer; however, only the release profiles of minitablets containing spray-dried lactose and coated with Eudragit to 12% weight gain were similar to the reference product [236]. Szczepanska et al. [237] conducted a study on the optimization of the enteric coating of pantoprazole minitablets with a diameter of 3 mm. Two fluid bed coaters (Aircoater 025 and 4M8-Trix) were used in the study. To obtain the cores of the minitablets, pantoprazole was granulated using sodium carbonate in a high-speed wet granulation process. Subsequently, the granules were tabletted with excipients using a rotary tableting machine. The resulting minitablets were subcoated with hypromellose and PEG 6000 solution (9:1), and then with Eudragit L 30 D 55. The study was based on the concepts of quality by design (QbD), design of experiments (DoE), and full factorial design to identify critical parameters of the film coating process. Four critical factors were identified, for which optimum values were then determined in each of the instruments used. These were inlet air temperature, product temperature, the flow rate of the coating mixture, and spraying pressure. It was observed that the values of the parameters studied are influenced by the apparatus used for coating. Although the study was conducted at the laboratory scale, the results can be used to predict problems during the scale-up process [237].

### 4.4. Pellets

Pellets are spherical-shaped granules with a size of 0.5–1.5 mm. They can be used to fill capsules or sachets, they can be compressed into multiparticulate tablets, or they can be administered in the form of oral suspension [238]. Two basic methods of pellets production include a coating of inert cores with the active substance or the formation of pellets from the mass containing the active substance (with extrusion-spheronization or granulation method). Both of these methods can be followed by the application of successive functional layers [239,240,241,242,243].

He et al. [239] compared the omeprazole release from the pellets produced with two different methods. Pellets with API in the core were prepared with the extrusion and spheronization method, while others were fluid-bed coated with a layer of omeprazole. The faster release occurred from pellets coated with omeprazole layers. After 6 min, the entire dose of API was released, while in the case of pellets with omeprazole inside the core, only 45% of it was released during this time. The researchers also evaluated the effect of the binding agent (PEG 6000, PVP K30, and mannitol) used in the coating solution on the release rate of omeprazole. However, the difference between them was insignificant [239].

An equally important aspect related to the delayed-release pellets is the effect of the coating process parameters and the excipients used on the functionality of the dosage form. In the case of pellets containing PPIs, the most commonly used method is fluid bed coating [30,31,238,239,244,245,246,247,248,249]. During this process, a solution or dispersion containing coating substances is sprayed with a nozzle onto a surface of pellets circulating in heated air. This method allows to form several layers of coatings with different functionalities. In the case of PPIs, a three-layer coating is usually formed, consisting of the drug substance, the neutral interlayer, and the enteric polymer coat. Such a design allows to achieve optimal stability of PPIs during manufacturing, as well as long-term storage of the drug. The neutral layer allows for the separation of the drug substance from the polymer containing acidic functional groups, which might negatively affect its stability. In the literature, HPMC or HPC were reported as polymers used to form the neutral sublayer, while methacrylates, e.g., Eudragit L, were often chosen as the enteric coating [238,244,245,246,247,248]. Han et al. [241] evaluated the effect of three types of Eudragit L solutions on the properties of pantoprazole pellets. They included an aqueous dispersion of Eudragit L100-55, an organic solution, and a 30% aqueous dispersion of Eudragit L 30D-55. The difference in the release profiles of the drug substance between the polymers was negligible. Nevertheless, API was released most rapidly from Eudragit L 30D-55-coated pellets, while it was released more slowly from pellets having organic coating. In terms of thermodynamic properties, solubility, or acid resistance, the coating from the organic solution of Eudragit L100-55 was shown to outperform the 30% aqueous polymer dispersion [241]. In another study, Fang et al. [249] described the development of pellets with an enteric coating that was a blend of Eudragit L and HPMCAS. It improved the stability and bioavailability of lansoprazole compared to non-blended polymeric coatings [249].

Tirpude et al. [250] conducted a study comparing the effectiveness of four types of enteric coated pellets with sodium rabeprazole. Eudragit L 30D-55 and hypromellose phthalate HP-55 were used as functional polymers. The pellets containing the active substance were first coated with a protective layer of HPMC. The first formulation was coated only with Eudragit L, another with a mixture of Eudragit L and NE 30D in a 9:1 ratio. The third formulation had a layer of HP-55, while the fourth had two layers: the first with Eudragit L and the second with HP-55. It was observed that the pellets coated with Eudragit had a smooth and glossy surface, in contrast to the pellets coated with a cellulose derivative whose surface was only smooth. Although all the formulations prepared met the delayed release criteria, pellets coated with Eudragit L have shown the poorest results, while double-layered pellets exhibited the highest performance [250].

Pellets containing PPIs can also be coated with four layers, where the alkalizing layer is added in between the drug and the neutral sublayers. The alkaline coating can be composed of salts, such as disodium hydrogen phosphate, sodium chloride, or sodium carbonate. Its function was to protect the API from decomposition in an acidic environment, and what is more, to increase the solubility of PPIs by promoting its conversion to an amorphous form [239]. He et al. [31] investigated the influence of the sodium bicarbonate content of the alkalizing layer on the stability of lansoprazole sodium in micropellets. The inert cores were coated with the drug layer to obtain 25% of the weight gain, and then with the alkaline layer, composed of sodium bicarbonate in 4% HPMC dispersion. Subsequently, the pellets were coated with protective and gastro-resistant layers. The addition of sodium bicarbonate to the composition of the pellets increased their stability in the simulated gastric fluid environment. Pellets with alkaline layers containing the highest amount of sodium bicarbonate released 1% of API within one hour of the acid resistance test, showing excellent gastro-resistance properties. However, the presence of sodium bicarbonate also delayed also release of lansoprazole in the simulated intestinal fluid. Nevertheless, it met the requirements of 80% lansoprazole released in 60 min of the study. During the study, the pellets without the alkaline layer changed color after three days of storage, while the particles containing sodium bicarbonate remained stable for more than 10 days. The observed effect was associated with the ability of sodium bicarbonate to form a protective barrier against the influence of moisture, which promotes the interaction of the drug substance with the acidic groups of the coating polymer [31].

Not only have delayed-release pellets with PPIs been developed, but also sustained-release pellets. Kan et al. [30] produced esomeprazole pellets coated with Eudragit RS and RL and Eudragit L-55. To avoid interactions, each layer of the coating was separated by a protective sublayer. The pellets were characterized by good resistance to acidic environments and prolonged release of the active substance for up to 10 h [30]. Gaudio et al. [251] developed beads of sodium alginate and SBA-15 with omeprazole, intended for the treatment of pediatric patients. SBA-15 is a mesoporous silicate that absorbs the therapeutic substance into its porous structure, which prevents its crystallization and agglomeration. Sodium alginate was used as a polymer to prolong release. The researchers used the gelation reaction of alginate under the influence of calcium ions. The obtained beads were characterized by a diameter of approximately 1.25 mm and sustained omeprazole release for up to 6 h [251].

Another interesting research was conducted by Hung et al. [252] who developed a method to obtain pulsatile pellets using rupturable controlled release membranes. Pellets containing a therapeutic substance and an osmotically active substance—sodium chloride—were fluid bed-coated with a dispersion of Eudragit RS/RL with triethyl citrate (TEC) and talc. The principle of this system was that the water entering the tablet, slowly generated osmotic pressure, which after reaching a critical value, would break the continuity of the membrane and cause the release of the therapeutic substance. The researchers tested three therapeutic substances (omeprazole, omeprazole sodium, and propranolol hydrochloride). In the case of omeprazole, it was not possible to generate adequate osmotic pressure because of the poor solubility of the substance. Omeprazole sodium, although much better soluble in water, also failed to achieve complete release from the system as a result of ionic interactions with Eudragit RS. Satisfactory results were achieved only for propranolol HCl, by modifying the thickness and composition of the coating [252].

### 4.5. Tablets

The largest number of publications related to the development of PPI dosage forms concern tablets. These include publications on delayed, prolonged, controlled, or pulsatile release tablets, as well as enteral, ODT, gastro-retentive, and MUPS tablets (Table 5).

Delayed-release tablets are manufactured using enteral polymers, among which the most commonly used are derivatives of methacrylates or cellulose (hydroxypropylmethylcellulose phthalate). As in the case of pellets, the literature provides examples of the manufacturing of an intermediate layer, which separates the enteric coat from the core with the therapeutic substance. HPMC or polyvinyl alcohol dispersions are used for this purpose [253,254,255,256,257,258,259,260,261]. The intermediate layer is intended to protect the therapeutic substance sensitive to acidic groups of enteral polymers [256]. Aravind et al. [262] proposed another solution to prolong the stability of pantoprazole. Researchers developed drug-resin complexes, which were compressed into tablets. The stability and content of the drug substance in the manufactured tablets were found to be similar to the reference delayed-release product [262].

During the compounding of the tablet, the compatibility of the excipients with the API needs to be carefully considered. Migoha et al. [263] showed that omeprazole can interact with components of the tablet mass. In preformulation studies, they observed that omeprazole interacted with Aerosil 200 (colloidal silica), turning dark purple or dark brown, depending on the test conditions. For other excipients, such as sodium lauryl sulfate, Avicel PH 101, magnesium stearate, lactose, and starch, no change was identified [263].

Various attempts have also been made to determine the effect of the enteral polymers used on the release rate of the API and the overall stability of the drug. Nair et al. [256] conducted a study in which they compared the properties of esomeprazole tablets coated with Eudragit L-30 D-55, hydroxypropylmethylcellulose phthalate, cellulose acetate phthalate, or Acryl-EZE (methacrylates). The tablet cores were coated with Opadry for up to 3% weight gain before applying the enteral coating. This amount was sufficient to protect the drug substance from interaction with the acidic residues of the enteral polymers. Enteric coating up to 5% weight gain was too little to prevent the release and degradation of esomeprazole during dissolution studies. However, the coating to 8% weight gain appeared to fulfill its function and ensured the stability of the formulation throughout the study period. Among the enteral polymers tested, methacrylate derivatives were found to provide the fastest API release at pH = 6.8 [256].

Bozdag et al. [264] reached the opposite conclusion, coating omeprazole tablets with enteral polymers for up to 4% weight gain. In this study, Eudragit S was used instead of Eudragit L. Only tablets coated with hydroxypropylmethyl cellulose phthalate and cellulose acetate phthalate were found to meet the criteria for delayed release according to USP XXII (not less than 75% of API released within 45 min in a phosphate buffer environment pH = 7.4). Although all the formulations produced were characterized by acid-resistance, they did not meet the accelerated stability test criteria [264]. To avoid the use of enteral polymers, Das et al. [265] and Lee et al. [33] proposed a tablet-in-tablet formulation based on the use of alkaline substances in the coating of the tablets To prepare the outer layer, sodium bicarbonate was compressed together with other excipients and with the core containing the therapeutic substance [33,265]. Rabeprazole release studies were carried out in phosphate buffer at pH = 8.0 and showed complete release within 30 min of the test. The amount of sodium bicarbonate needed to neutralize hydrochloric acid was calculated based on the acid neutralization capacity method (USP Pharmacopeia) and was estimated at 800 mg. Studies on beagle dogs have also been conducted in which the maximum concentration of rabeprazole was reached three times faster compared to the reference product with similar AUC values [33].

An innovative approach to manufacturing delayed-release tablets containing PPIs was proposed by Alsulays et al. [266]. They used a hot melt extrusion process to improve the physicochemical properties of lansoprazole. Kollidon VR 12 PF was used as a matrix polymer, magnesium oxide as a stabilizer, and Lutrol VR F 68 as a plasticizer to produce the extrudate. The process was carried out at a reduced temperature (65 °C) to protect the drug substance from degradation. The lansoprazole content in the optimized extrudate was 97%. The extrudate was tableted and coated with Eudragit L100-55 to 9% weight gain. The manufactured tablets met the delayed release criteria, releasing less than 10% of lansoprazole when tested in hydrochloric acid and more than 80% of the substance when tested in the simulated intestinal fluids. Compared to the pure substance, the release of lansoprazole from the extrudate tablets was faster, due to the substance’s transition into an amorphous form under the extrusion process. The tablets were stable for at least 6 months at 25 °C/60% RH [266].

A method of production of tablets containing esomeprazole [267] for colon delivery was also developed. In the first stage, the tablet core containing the therapeutic substance was produced by direct compression. The tablet core was then surrounded by a protective layer consisting of a mixture of ethylcellulose and HPMC powders. In the final stage, the bilayer tablet was coated with a solution of hydroxypropylmethylcellulose phthalate (HP-55). Tablets were characterized by acid resistance with further release of therapeutic substances for up to 8 h [267].

In order to achieve a prolonged gastric acid suppression effect, attempts have been made to manufacture extended-release tablets of PPIs [268,269,270,271,272].

Divya et al. [268] conducted an extensive study in which they tested different types of polymers and their concentrations for the preparation of tenatoprazole extended-release matrix tablets. The tablets were prepared by direct compression of a pre-granulated tablet mass containing polymers. The polymers tested included Carbopol 941P and 974P-NF, as well as Eudragit L 100, S 100, RS 100 and RL 100 comprising from 16 to 32% of the tablet mass. Among the formulations produced, tablets containing 16% and 32% Carbopol 941P and 24% Carbopol 974P-NF had sufficient resistance to hydrochloric acid and released the drug substance for up to 12 h [268]. The same research group developed other extended-release tablets that contained ilaprazole as an active substance. Core tablets consisting of polymers that form an extended-release matrix with the drug substance were coated with a protective layer of HPMC for a weight gain of 2%. The tablets were then coated with Acryl-EZE-80 enteral polymer to gain 8%, 12% and 16% weight. Although the release of ilaprazole was achieved up to 12 h, the tablets had insufficient gastric resistance [271]. Wilson et al. [269] conducted a similar study for pantoprazole, in which HPMC, cassava starch and PVP in amounts of 20–100 mg per tablet were tested as extended-release excipients. Cellulose acetate phthalate (CAP) and Eudragit L100 were chosen as enteral polymers. The CAP-coated tablets appeared to fully prevent the release of API during acid-resistance tests. HPMC tablets were shown to have the best release profiles [269]. The development of extended-release tablets for rabeprazole [270] and dexlansoprazole has also been described [272].

Pulsatile tablets have also been developed for proton pump inhibitors. This solution is applicable to therapeutic substances that should be used according to the principles of chronotherapy. In the case of PPIs, these systems could be administered before going to bed to avoid nocturnal acid breakthrough or to avoid the need to take the drug in the morning. Arora et al. [273] developed a pulsed-release tablet of pantoprazole based on modifications to the composition of the coating of the tablets. They used the ethylcellulose and HPMC polymers in a 1.5:1 ratio. This composition allowed achieving a delayed release of the drug substance by 3 h, followed by a rapid burst release of the substance within an hour [273].

Other forms of tablets intended for the administration of PPIs include multiparticulate unit tablet systems, among them MUPS tablets (multiple-unit pellet systems) and orodispersible tablets (ODTs). Preparation of such tablets with PPIs is based on the compression of the modified-release pellets, or other multiparticulates together with excipients into the final form of the tablet. In the case of ODTs, important excipients include superdisintegrants, essential for quick disintegration of tablets in the mouth, upon contact with saliva. The advantages of multiparticulate unit tablets include rapid passage through the gastrointestinal tract and a much lower risk of loss of drug substance due to damage to the enteral coating than in the case of conventional tablets. Sonar et al. [274] have developed MUPS tablets for pantoprazole. They consisted of pellets, the core of which was coated with the therapeutic substance. The subsequent layers consisted of a protective layer of HPMC, an enteric coating of Eudragit L 30D-55 and PlasAcryl, covered by another protective layer of HPMC and PEG 6000. Tablets met the delayed release criterion, were resistant to hydrochloric acid, and the release of 80% of the therapeutic substance in phosphate buffer occurred in less than 45 min [274]. A study by Al-Zidan et al. [275] described the effect of excipients used for the preparation of tablets with esomeprazole on their properties. Tablets containing dibasic calcium phosphate as a diluent was characterized by high friability (more than 1%), which eliminated this formulation from further studies. Tablets containing MCC were chosen as the optimal formulation because of the additional effect of wrapping the pellets and protecting them from mechanical damage. Furthermore, these tablets proved to be the hardest despite the application of a low compression force, due to the plastic transformation of MCC during compression. The excipients were mixed with the pellets in a 1:1 ratio, because then no segregation of the tablet mass was observed [275].

A lot of attention has also been paid to the development and improvement of orodispersible tablets (ODTs) containing PPIs. Shimizu et al. [276,277,278] published three papers describing the manufacture of lansoprazole ODTs. The tablets developed by the researchers contained two types of granules: enteric-coated with therapeutic substance and inactive granules, designed to give the formulation a pleasant taste [276,277,278]. The first study evaluated the effect of compression force on the resistance and release profile of the manufactured enteric granules. It was observed that methacrylate pellets manufactured without the addition of a plasticizer were too fragile to ensure formulation stability during release studies. The addition of talc to the coating dispersion also contributed to this effect. Therefore, an optimized formulation containing 20% triethyl citrate (TEC) as a plasticizer was developed. It was determined that a 9:1 ratio of the methacrylic acid copolymer dispersion to the ethyl acrylate-methyl methacrylate copolymer dispersion provided adequate strength and flexibility to the enteric coat. Instead of talc, 5% glyceryl monostearate was used to avoid coating breakage. In vitro and in vivo studies showed similarities in the release profiles of the manufactured pellets and the reference capsule formulation [276]. In a subsequent series of studies, it was shown that enteric-coated granules containing TEC have an unpleasant, bitter taste. Furthermore, stability studies showed that during storage, TEC is incompatible with lansoprazole. To solve this problem, an improved formulation of granules coated with seven layers was proposed. The core of the tablets was coated with the drug substance, which was protected by a layer of HPMC. The next three layers contained the enteral polymers mentioned above, except that the first and third layers contained PEG 6000 instead of TEC, which was compatible with lansoprazole. Between them was the actual enteral coating with TEC added. A final mannitol coating layer was added to improve the taste of the formulation. This complex approach ensured both optimal stability and a pleasant taste of the enteral pellets [277]. The third publication describes the method for manufacturing ODTs with lansoprazole. The formulation included enteric-coated pellets and placebo pellets. Inactive granules included microcrystalline cellulose, mannitol, and low-substituted hydroxypropyl cellulose (L-HPC). Optimal dissolution parameters and a disintegration time of less than 30 s were obtained in the case of a formulation containing 47.5% coated granules. It was also pointed out that L-HPC improves palatability and has no rough texture compared to other excipients, as a result of its reduced water absorption. Therefore, it can be used successfully in ODT tablets as a binder and disintegrant [278]. In the study by Choursiya et al. [279] cyclodextrins were used to improve the solubility of lansoprazole in orodispersible tablets [279].

Since lansoprazole was the first PPI available in an ODT form, many publications considered its bioavailability from this form of the drug. In clinical studies, the bioavailability of lansoprazole orodispersible tablets at doses of 15 and 30 mg has been shown to be comparable to capsules containing enteric coated pellets [280,281,282,283]. Furthermore, Chono et al. [284] conducted a study on the ingestibility, acceptability, palatability and quality of lansoprazole ODT products available on the Japanese market [284]. A separate study also evaluated the physical properties of these formulations, such as wetting time or tablet strength [285].

Examples of ODT formulations with omeprazole [286,287] were also reported in the literature. Sai et al. [288] investigated the properties of ODT with pantoprazole, differing in the superdisintegrant used. They compared formulations containing crospovidone or sodium starch glycolate (Primojel) at three concentrations of 2.5%, 5% and 10%. It was observed that as the concentration of superdeintegrants decreased, the disintegration time of the tablets increased. Tablets containing 10% crospovidone had the fastest dispersion time of 8 s [288]. These results confirm those obtained by Balamuralidhara et al. [289], who conducted a study comparing 5 disintegrants at concentrations of 5% and 10% used to produce ODTs with rabeprazole. The fastest disintegration time was obtained for the formulation containing 10% crospovidone [289]. Moreover, an innovative method for producing ODTs with omeprazole was developed by Alhusban et al. [286]. Tablets containing gelatin, carrageenan, and alanine in various concentrations were produced using a freeze-drying method. Researchers developed a mathematical model that allowed them to accurately predict the relationship between the applied concentrations of selected formulation components and the properties of the produced formulations, i.e., disintegration time, hardness, viscosity, and pH [286].

### 4.6. Fixed-Dose Combination Products

Vynckier et al. [290] investigated a method to produce a fixed-dose combination product by hot-melt co-extrusion. Researchers attempted to produce a core containing naproxen, which was then coated with a layer containing immediate-release esomeprazole magnesium. Eudragit L100-55, hydroxypropylmethylcellulose acetate succinate (HPMC-AS-LF), and hydroxypropylmethylcellulose phthalate (HPMCP-HP-50) were tested as core-forming polymers. Depending on the polymer selected, different extrusion temperature ranges and naproxen load (15–50%) were tested. Triethyl citrate was used as a plasticizer. Formulations made from HPMC-AS-LF with the addition of TEC showed the best release profiles. All of these formulations remained stable in hydrochloric acid. During dissolution studies carried out in phosphate buffer of pH = 6.8, the highest amount of naproxen was released from formulations containing 15% API. Extrudates containing a 30% drug load released esomeprazole at the slowest rate. In the case of the formulation containing esomeprazole magnesium, extrudates based on Eudragit L100-55, HPMC-AS-LF, and HPMCP-HP-50 could not be produced due to complete degradation of the drug. However, an immediate-release drug layer composed of polyethylene oxide 100 K and PEG 4000 in a 1:1 ratio was developed, in which esomeprazole was stable. After the co-extrusion process, it was found that the layer containing esomeprazole interacted with the naproxen core, causing the discoloration of the extrudate. It was concluded that to make the process effective, an additional insulating layer must be applied between the core and the coating layer. Therefore, hot-melt extrusion seems to be an interesting, albeit very challenging direction in the case of PPIs [290].

### 4.7. Bilayer Tablets

The bilayer tablets contain two therapeutic substances that are located in separate layers of the tablet. Therefore, it is possible to administer two drugs for the same disease in a single tablet, even though the therapeutic substances are incompatible. For example, a combination of esomeprazole with clarithromycin [291,292] and amoxicillin with lansoprazole [293] dedicated for the treatment of H. pylori, as well as a combination of esomeprazole with aceclofenac [294] has been proposed.

Israr et al. [291] developed bilayer floating tablets containing 20 mg of esomeprazole and 250 mg of clarithromycin with sustained release of therapeutic substances. The bilayer formulation was based on Eudragit RS forming an insoluble matrix. In addition, sodium bicarbonate was used as a gas-generating agent. Tablets were prepared by direct compression. Initially, the cavity of the die was filled with a tablet mass containing clarithromycin and compressed. The die cavity was then again filled with a tablet mass containing esomeprazole and compressed to the final form of a bilayer tablet. Due to the use of Carbopol 934, the tablets had a high swelling index and, therefore, good floating properties. The total flotation time was 24 h, with a lag time of as little as 25 s. The content of API in individual formulations was above 98.89%. Dissolution studies were conducted for 6 h in 0.1 M hydrochloric acid and for 18 h in phosphate buffer of pH = 7.4. Approximately 45–55% of esomeprazole was released in acid, depending on the formulation. The dissolution profiles were compared with the profiles of reference drugs with immediate release and no similarities were detected between them [291].

Singh et al. [293] developed bilayer tablets with immediate-release of lansoprazole and extended-release of amoxicillin. The amounts of therapeutic substances used were 15 mg and 500 mg, respectively. To prepare the immediate-release layer, the components of the tablet mass, including PVP K30 and sodium starch glycolate in three different concentrations, were granulated by wet granulation. The same process was used for the tablet mass containing amoxicillin and the sustained-release polymers: ethylcellulose, PVP, and HPMC K100. Initially, the granules were compressed separately to optimize their composition. Among the formulations of lansoprazole, the granules containing the highest concentration (8%) of sodium starch glycolate had the fastest API release. In the case of amoxicillin-containing granules, those with ethylcellulose and HPMC in a ratio of 1 + 3 were selected. Subsequently, the granules were compressed into layers to obtain a tablet with a diameter of 4.7 mm. The content of the individual substances was 97.46% and 98.25% for lansoprazole and amoxicillin, respectively. Dissolution studies of bilayer tablets were carried out for 2 h in 0.1 M hydrochloric acid and then for 8 h in a phosphate buffer (pH = 6.8). The complete release of lansoprazole occurred after 15 min of the study, while 85% of amoxicillin was released after 10 h [293].

Nijhu et al. [294] also attempted to produce a bilayer floating tablet. The immediate-release outer layer contained esomeprazole, sodium starch glycolate as a superdisintegrant, and gas-generating agents. The inner layer contained aceclofenac, which had delayed release due to the addition of one of three types of HPMC or xanthan gum. A six-hour flotation time was achieved in most of the batches produced, except for one formulation containing 40% xanthan gum that did not exhibit flotation properties [294]. However, in all the studies described, susceptibility to the degradation of PPIs in the acidic environment of the stomach was not mentioned. Therefore, their degradation under in vivo conditions must be considered in further studies, particularly because all of them showed immediate release of the therapeutic substance.

### 4.8. Floating Tablets

Among articles on the drug formulation of PPIs, several papers focus on the development of floating tablets. The main goal of the manufacture of floating tablets is to achieve a prolonged or controlled release of the drug substance. Due to the presence of an alkaline substance, i.e., sodium bicarbonate in the tablet formulation, they temporarily increase the pH of gastric juice. This feature can be used for immediate relief from GERD symptoms, in addition to protecting PPIs from disintegration in the stomach.

Abbas et al. [295] developed floating tablets with pantoprazole sodium using pectin as a raft-forming polymer. Pectin is a natural polysaccharide that forms a gel by undergoing the cross-linking reaction in a gastric environment induced by calcium cations. The pectin tablets were formulated with sodium bicarbonate and citric acid as gas-generating excipients, to achieve fast disintegration time and promote the formation of rafts on the surface of gastric fluids. To increase the viscosity of a gel and to provide sustained release of pantoprazole sodium, HPMC was used. Calcium bicarbonate was added to the formulation to improve the stability of the rafts. Disintegration studies showed that the time required for forming a raft was up to 97 s. The disintegration time of the tablets depended on the amount of pectin and sodium carbonate in the tablet formulation. Pantoprazole sodium has been released for 8 h while the tablet floated on the surface of the simulated gastric fluid for 24 h [295].

The floating tablets exhibiting prolonged release of lansoprazole have also been described by Bindu et al. [296]. The investigated formulations contained HPMC as a gelling agent and sodium bicarbonate as a gas-generating excipient. The floating time of all tablet formulations was more than 24 h. Along with increasing HPMC content, the amount of drug substance released decreased. However, tablets containing the highest amount of HPMC (50% of a tablet mass), had the highest swelling index value [296]. A study by Reddy et al. [297] compared the effects of HPMC and sodium alginate content in tablet mass on pantoprazole release. Floating tablets containing a mixture of both substances in different ratios and one of the polymers were prepared. No significant differences were observed between the release profiles of the manufactured tablet batches. The optimized formulation containing only HPMC was characterized by a floating lag time of 60 s, a prolonged release of the drug substance of up to 8 h, and excellent stability during storage in climate chambers for one month [297].

Another example of the floating formulation are rafts containing rabeprazole sodium developed by Shah et al. [298]. In contrast to previously described formulations, this system is designed for the immediate release of the therapeutic substance. The formulations selected for the in vitro and in vivo studies were composed of 20 mg of rabeprazole, macrogol 400, mannitol, sodium bicarbonate, calcium carbonate, citric acid, and citrus pectin. The time needed for the neutralization of the simulated gastric fluid (SGF) was 46 min. Dissolution studies were conducted in four different acidic media: 0.1 M HCl, 0.5 M HCl, 1.0 M HCl, and SGF in a volume of 50 mL with sink conditions maintained. It was shown that 95% of the therapeutic substance was released within 20 min of the selected formulation regardless of the kind of medium used. The formulation was effective in maintaining the gastric pH above 3.5 to prevent reflux. The results of in vivo studies in albino rats proved that the developed raft-forming tablets provided a higher bioavailability of rabeprazole compared to the commercial capsule formulation [298].

Sonam et al. [299] developed extended-release gastro-retentive tablets of lansoprazole. Two types of polymers were tested: xanthan gum and gellan gum, which were added in the amount of 50–140 mg per tablet. Sodium bicarbonate and citric acid were added to the tablet mass to provide the gastroretentive properties of the formulation. Dissolution studies conducted in a 0.1 M hydrochloric acid exhibited complete release of lansoprazole during a 12-h study in the case of a formulation containing gellan and xanthan gum in equal amounts of 50 mg [299]. Nonetheless, the studies mentioned above are questionable in terms of the stability of PPIs released into a low pH environment.

### 4.9. Hydrogel Formulations

Another approach in the design of dosage forms with PPIs is the formulation of hydrogels. Like floating tablets, they contain a polymer that forms a dense matrix through which the drug diffuses. This dosage form has become increasingly attractive in recent years. Hydrogels have high biocompatibility and protect the drug from degradation due to their cross-linked structure. They can be used to achieve modified drug release or topical action. Taking into account these properties of hydrogels, several solutions have been proposed for the formulation of dosage forms with PPIs [300].

In the study by Saruchi et al. [301] a hydrogel composed of tragacanth gum and acrylic acid was used to control the pantoprazole sodium release. To obtain the drug-loaded matrix, the hydrogel was incubated in 0.2% pantoprazole sodium solution for 24 h, then washed and dried. The amount of pantoprazole released was measured at three pH values: 2.0, 7.0, and 9.2. After 30 h of study, the highest concentration of API was measured in the alkaline solution. The investigated hydrogel was found to be a sufficient carrier for the prolonged colonic release of the drug substance [301].

Gupta et al. [302] proposed the synthesis of a super porous hydrogel of pantoprazole sodium using acrylamide and methacrylic acid as crosslinking polymers. Each tested sample contained 20 mg of pantoprazole. Methylene bis-acrylamide was used as a crosslinking agent. The superporous gel structure was obtained by the addition of sodium bicarbonate, which was a source of foam-forming carbon dioxide. Other components of the mixture were crosslinking initiators and stabilizing agents (Ac-Di-Sol and Pluronic). To saturate the hydrogel with pantoprazole, the hydrogel was initially placed in a buffer for complete swelling and then transferred to the API solution. After complete absorption of the drug substance, the hydrogel was dried. The total API content of the formulations produced was 95–98%. In the swelling index test, the sensitivity of the hydrogel to the pH of the solution was observed. In an acidic environment, due to the presence of methacrylates, a slight increase in the swelling index was observed, while in the phosphate buffer at pH = 7.4 the volume of a hydrogel increased significantly. The time of the release of pantoprazole from the hydrogel ranged from 270 to 480 min and depended on the amount of methacrylate used, the degree of crosslinking of the polymer and the amount of stabilizing agents [302].

The paper by Sudhakaran et al. [303] describes a method for producing a floating hydrogel intended for prolonged pantoprazole release. The formulations developed contained sodium alginate or gellan gum as a crosslinking polymer. Both polymers were crosslinked in the presence of calcium ions and formed a dense matrix. To test the gelling ability of the formulations, 1 mL samples of polymer solutions were placed in 5 mL of 0.1 M hydrochloric acid. All formulations have a flotation time longer than 12 h. The content of drug substances in the samples ranged from 75.36% to 87.69%. In dissolution studies conducted in 0.1 M hydrochloric acid, the amount of pantoprazole released was from 77.80% to 87.12%. It was proved in fluorescence imaging studies conducted on mice that the formulations were effective in the gelation process and the hydrogel was presented in the stomach up to 6 h after administration [303].

Kumar et al. [304] described a method for producing hydrogel beads characterized by prolonged rabeprazole release. Beads were obtained by ionotropic gelation, where a solution of the drug and sodium alginate was dropped into a solution of calcium chloride. The obtained particles were coated with Eudragit S100 solution. In the dissolution test carried out in 0.1 M hydrochloric acid, the amount of rabeprazole released did not exceed 10%. The remaining active substance was released in a buffer stage for up to 8 h by diffusion mechanism. The higher the concentration of calcium chloride used to crosslink the alginate, the more compact the structure of the beads and the slower the release of the therapeutic substance was [304].

### 4.10. Mucoadhesive Tablets

Mucoadhesive tablets are another example of a polymer-based dosage form. They are convenient formulations used when the local or prolonged action of the API is required. An example of a condition in which they can be used is a duodenal ulcer.

Choudhury et al. [305] developed a formulation of mucoadhesive tablets containing 20 mg of omeprazole. Hydroxypropylmethylcellulose K4M, sodium carboxymethylcellulose, Carbopol-934P, and ethylcellulose were used as mucoadhesive polymers. Omeprazole pellets were compressed with excipients and then coated with a 0.5% solution of one of the mucoadhesive polymers. Subsequently, the tablets were coated with Eudragit L. The formulation containing HPMC was excluded from further studies due to inconsistency in process parameters. Obtained tablets were characterized by satisfactory friability, hardness, mass uniformity, and API content (89.7–91.3%). The formulation containing Carbopol 934P had the highest mucoadhesive strength (30 gm) and the highest swelling index value (1.51%). In contrast, tablets containing ethylcellulose were characterized by the lowest values of the parameters mentioned above. Dissolution studies were conducted in 0.1 M hydrochloric acid for two hours and in phosphate buffer (pH = 6.8 and 7.5) for 12 h. The amount of omeprazole released in hydrochloric acid from the tested formulations ranged from 12% to 17%. In the buffer stage, within 12 hours, 92% of omeprazole was released from tablets containing Carbopol 934P, and 67% from tablets with ethylcellulose. In the case of tablets containing sodium carboxymethylcellulose, 95% of omeprazole was released within 8–10 h. It was observed that the mechanism of drug release in each formulation was diffusion. Enteric-coated tablets containing Carbopol 934P were found to have the greatest implementation potential [305].

In another study, Reddy et al. [306] developed extended-release mucoadhesive tablets with pantoprazole. Tablets were prepared by direct compression of a tablet mass containing HPMC K4M, Carbopol 940NF and guar gum, respectively. Formulations containing 20, 30 and 40 mg of each polymer were prepared. The formulation containing 40 mg of HPMC had the best mucoadhesive properties. In addition, these tablets appeared to release the highest amount of drug substance (97% in 10 h) [306].

Mucoadhesive buccal drug dosage forms are used to protect the drug substance from degradation during passage through the gastrointestinal tract and to increase its bioavailability. When using this dosage form, the therapeutic substance is absorbed directly into the bloodstream through the oral mucosa, bypassing the hepatic first-pass effect and the unfavorable conditions of the gastrointestinal tract. Nonetheless, designing a buccal dosage form for PPIs presents quite a few challenges due to their instability in the slightly acidic or even neutral environment of the oral cavity. Furthermore, because PPIs are used in an inactive, unprotonated form, skipping the first-pass effect seems to be an unwelcome outcome when immediate drug action is expected.

Choi et al. [307] developed buccal tablets containing omeprazole, the stability of which they tested in human saliva. Tablets contained 20 mg of omeprazole. The different batches varied in the content of bioadhesive polymers, i.e., sodium alginate, HPMC, and the alkalizing substances: magnesium oxide, potassium phosphate monobasic, sodium phosphate monobasic, and sodium phosphate dibasic. The stability of the manufactured tablets was tested by placing them in 5 mL of human saliva for 4 h. Then, their appearance was evaluated and omeprazole content was determined. It turned out that magnesium oxide had the best properties to stabilize omeprazole in saliva. Additionally, it was found that with decreasing sodium alginate—magnesium oxide ratio and increasing HPMC content, the adhesion capacity of the tablets decreased. The researchers also determined that the amount of omeprazole absorbed within 15 min from the solution kept in the mouth of healthy volunteers was as high as 23% [307].

In the complementary study of the same research group, the effect of the croscarmellose sodium content on the rate of omeprazole release from buccal tablets was examined [308]. It was found that the addition of this excipient increased API release from the tablet by forming micropores in the tablet matrix and provided zero-order kinetics. However, too much croscarmellose sodium reduced the bioadhesive properties of the formulation. The optimized formula contained omeprazole, sodium alginate, HPMC, magnesium oxide, and croscarmellose sodium in the ratio 10:12:3:25:5. The study was supplemented with pharmacokinetic data from hamster studies. It was found that omeprazole had been absorbed in approximately 13% of the manufactured buccal tablets, while the maximum concentration was determined after 45 min and remained between 146 and 366 ng/mL for up to 6 h [308]. Yong et al. [309] also reported that the addition of croscarmellose sodium accelerates the release of omeprazole from buccal tablets but negatively affects its stability in human saliva. Therefore, alkaline substances such as magnesium oxide, sodium/potassium phosphate monobasic, and sodium phosphate dibasic were added to the tablet mass. It was shown that the addition of magnesium oxide in the amount of 50% of the tablet mass provides optimal stability of omeprazole for 4 h, assessed as the absence of a change in tablet color from white to dark in contact with saliva [309].

Attempts have also been made to develop buccal films containing PPIs that can be used to treat pediatric patients. Mucoadhesive films are an attractive alternative to conventional tablets, because of the lower risk of choking. Khan et al. [310] attempted to produce buccal films based on various polymers, i.e., hydroxypropylmethylcellulose, methylcellulose, sodium alginate, carrageenan, or metolose. The films were prepared by dissolving the polymers (1% *w*/*w*) in aqueous and ethanol-water solutions (10% and 20% ethanol). The other components of the prepared gel were omeprazole, PEG 400 as a plasticizer, and L-arginine as a stabilizer. After drying, the films were subjected to a morphological evaluation. Among the formulations produced, films made from metolose in a solution containing 20% ethanol, 0.5% PEG 400, omeprazole and L-arginine in a 1:2 ratio had the best properties. These films provided optimal stability for omeprazole and were characterized by the best transparency, flexibility, ease of detachment from the Petri dish and molecular dispersion of the drug substance [310].

### 4.11. Oral Liquid Suspensions

Another well-known method of administration of PPIs and other poorly water-soluble drugs is their application in the form of suspensions. They are used mainly in the treatment of pediatric and critically ill patients who require drug administration via feeding tube. In numerous hospital pharmacies, PPIs are suspended in an 8.4% sodium bicarbonate solution. Boscolo et al. [311] investigated the stability of suspensions prepared in this way, with the addition of appropriate excipients. One tested formulation included crushed pellets containing omeprazole, while the second one powdered omeprazole. Both had an API concentration of 2 mg/mL. Additional ingredients used for the preparation of suspensions included sodium carboxymethyl cellulose as a suspending agent, sodium bisulfide as an antioxidant, and humectants in the form of glycerin and sorbitol. In addition, sweetening and flavoring agents were added to mask the bitter taste of omeprazole. The suspension of omeprazole pellets and powder was found to be stable at 4 °C for 150 and 90 days, respectively. When stored at room temperature, the suspension with pellets was stable for 14 days, while the one with powdered omeprazole was stable only for 1 day [311].

A different approach to preparing PPI-containing suspensions is to produce nano- or microparticles with modified release properties. They can be suspended in a vehicle other than sodium bicarbonate solution, which eliminates potential problems associated with excessive sodium intake. Ronchi et al. [312] developed delayed-release micropellets dedicated to administration in the form of syrup. The microcrystalline cellulose pellets were coated with 5 layers to achieve a stable formulation. Initially, the pellets were coated with a layer containing a drug substance, followed by an insulating layer made of PVP. The third layer obtained from an ethanolic solution of Eudragit L-55 was the actual enteric coat. The pellets were then coated with an insulating layer and finally with an outer layer of Eudragit E100, soluble in the gastric environment. The diameter of the pellets was about 500 µm. The coatings used for the formulation protected omeprazole from gastric acid, while in a medium of pH = 6.8, the complete release of API occurred within 45 min. The prepared pellets were stable during storage at room temperature in syrup for 10 days [312]. In another study, Diefenthaeler et al. [313] described the manufacture of nanoparticles with omeprazole using an emulsification method. Eudragit RS 100 was the matrix-forming polymer, while Eudragit L-55 forms an enteric coating. The nanoparticles had a diameter of 174 nm. When the enteric polymer was used in the amount of 0.03%, up to 11% of the therapeutic substance was released during a 1 h dissolution test in a hydrochloric acid. Nevertheless, the nanoparticles had the ability to prevent gastric ulcer formation in mice. With slight formulation modifications, the proposed nanoparticles have future potential to be used in the treatment of pediatric patients [313].

### 4.12. Transdermal Delivery

The advantages of transdermal administration include increased patient compliance and improved bioavailability of the drug, due to bypassing the gastrointestinal tract and the first-pass effect. Attempts have been made to produce this form of the drug for a group of PPIs as well.

Soral et al. [314] aimed to develop transdermal patches containing rabeprazole sodium. Hydroxypropyl cellulose (HPC), polyvinyl pyrrolidone K-30, and polyvinyl pyrrolidone K-90 were tested as the polymers that form the patch. The polymers were dispersed in ethanol and then PEG 400 plasticizer and Tween-80 or azone were added as absorption enhancers. The dispersions were poured onto a dish lined with aluminum foil that was used as a backing membrane. Dissolution studies were conducted in a dissolution apparatus V according to USP in phosphate-buffered saline (PBS) of pH = 7.4. About 60% rabeprazole was released from the formulation containing 5% HPC. It was observed that the amount released was lower with the higher the concentration of the polymer used. In studies of snakeskin permeability, the highest values of this parameter were found in the formulation containing 5% HPC and the absorption enhancer-Tween 80. It was estimated that if the patch had an area of 16 cm^2^ it could meet the therapeutic requirements [314].

A different approach to the manufacturing of transdermal patches was described by Lin et al. [315]. Researchers produced nanostructured lipid carriers containing lansoprazole, which subsequently were suspended in a hydrogel. The lipid nanoparticles were obtained by dissolving glyceryl monostearate and stearylamine in the appropriate weight ratio. Stearylamine was used to increase the stability of API. Lansoprazole dissolved in methanol was then added to the mixture. The organic phase was suspended in water at 65 °C with the addition of solubilizers (SDS and Pluronic F66). The emulsion was sonicated and cooled. The produced nanoparticles were mixed with hydrogels containing 1% xanthan gum, 5% glycerin, and penetration enhancers isopropyl myristate or menthol. Based on in vitro and in vivo studies, it was concluded that the formulation allowed to maintain therapeutic concentrations of lansoprazole for at least 24 h [315].

Less optimistic data were presented by Haas et al. [316], who conducted a clinical study on the bioavailability of omeprazole after transdermal administration. The tested formulation was a pleuronic lecithin organogel (PLO) with an omeprazole concentration of 50 mg/mL. An amount of gel equivalent to 40 mg of the drug substance was administered to the skin of the forearm without an occlusive dressing to 8 healthy volunteers. It was shown that the transdermal form of the drug was not bioequivalent to the oral capsule and that the permeability of the drug through the skin was poor [316].

### 4.13. Suppositories

Suppositories are a form of drug used in both children and adults. It is particularly dedicated to newborns, for whom the administration of oral forms of the drug can be problematic, e.g., due to the risk of damaging the enteric coating. Bestebreurtje et al. [317] developed suppositories containing omeprazole intended for newborns. As reported, the dose of omeprazole for infants is 0.5–3.0 mg/kg/day. Consequently, the researchers developed formulations containing omeprazole at 3–10 mg/suppository. Witepsol H15 was used as the base to which 100 mg of L-arginine was added to prevent the degradation of omeprazole during storage. To prepare suppositories, the base was melted at 50–60 °C and cooled to about 35 °C. L-arginine was micronized in a mortar, mixed with an omeprazole, and added to the melted base. Suppositories containing the lowest and highest doses of API were subjected to stability tests in a climate chamber at 4 °C, and in a dark and lighted room below 25 °C for one year. The suppositories appeared to change color and texture during storage when exposed to light but remained stable in a dark room [317]. The safety and efficacy of omeprazole suppositories were demonstrated in clinical trials in a group of 17 infants. The efficacy of omeprazole administered in oral and rectal forms was similar [318].

### 4.14. Intravenous Formulations

Attempts have also been made to increase the stability of proton pump inhibitors in intravenous solutions. Holvoet et al. [319] developed a powder for injection or infusion containing cyclodextrins. It was observed that omeprazole had better solubility in solutions containing 40% hydroxypropyl-β-cyclodextrins than in plain buffer solutions at pH in the range of 7.4–10. It was also found that the higher the pH of the solution, the higher the complexation efficiency of omeprazole with cyclodextrins. Two formulations containing cyclodextrins in a concentration of 40% prepared at pH = 11 and pH = 12 were used, as those with the greatest implementation potential. Nevertheless, the dissolution rate of omeprazole in the lower pH solution was much longer than at pH = 12 (45 min vs. 15 min), which is of practical importance in the preparation process. Both selected formulations, after reconstitution, yielded solutions for injection at physiologically acceptable pH [319].

Möschwitzer et al. [320] attempted to produce a nanosuspension with omeprazole using DissoCubes^®^ technology. To obtain the suspension, omeprazole powder was dispersed in a solution of 8.4% sodium bicarbonate and 1% poloxamer. The dispersion was then homogenized. When the process was carried out at room temperature, discoloration of the suspension was observed due to the decomposition of omeprazole. The nanosuspensions produced at 0 °C were stable within 3 days after preparation. To determine the chemical stability of omeprazole, prepared nanosuspensions and a solution of 8.4% sodium bicarbonate and omeprazole were stored at 4 °C for one month. No significant changes were observed in the omeprazole content during storage of the nanosuspensions, in contrast to the solution, in which the omeprazole content gradually decreased [320].
pharmaceutics-14-02043-t005_Table 5Table 5PPIs’ formulations described in the literature.FormulationPPIDevelopment StageDescriptionReferencesNanoparticlesOmeprazoleIn vitro In vivo antiulcer activity (rats)Enteric-coated nanocapsules,Functional polymers: HPMCP, PVAP,Obtained by emulsification method[187]In vitroIn vivo antiulcer activity (rats)Enteric-coated nanoparticles,Functional polymers: Eudragit L 100–55, chitosan,Obtained by complex coacervation method[190]PantoprazoleIn vitropH-sensitive polymeric nanoparticles,Functional polymers: Eudragit S 100, HPMCP (HP–55),Obtained by nanoprecipitation method[189]In vitroSustained-release solid lipid nanoparticles (SLNs),Functional polymers: ethylcellulose, chitosan, HPMC K100, PVA,Obtained by nanoprecipitation method[191]Pantoprazole + AceclofenacIn vitroIn vivo (rats)Sustained-release nanofibers,Functional polymers: zein, Eudragit S 100,Obtained by single nozzle electrospinning[193]LansoprazoleIn vitroSustained-release nanoparticles,Functional polymers: Eudragit RS 100,Obtained by oil-in-water emulsion-solvent evaporation method[188]In vitroControlled-release nanosponges,Functional polymers: ethylcellulose, PVA,Obtained by emulsion solvent diffusion method[192]In vitroImmediate-release niosomes,Obtained by reverse phase evaporation method[194]In vitroNanosuspensions composed of β-cyclodextrin-API complexes or β -cyclodextrin-API nanosponges,Obtained by physical method or polymer condensation method respectively[196]Lansoprazole + curcuminIn vitroBioactive solid self-nanoemulsifying drug delivery systems (Bio-SSNEDDS),Functional excipients: black seed oil, Zanthoxylum rhetsa oil,Obtained by emulsification method[197]EsomeprazoleIn vitroEx vivo permeability studyIn vivo PK and PD studies (rats)Proniosomes,Functional excipients: maltodextrin (carrier), cholesterol,Obtained by slurry method[195]MicroparticlesOmeprazoleIn vitroIn vivo PK study (rabbits)β-cyclodextrin-API complexes encapsulated with antacids in gelatine capsule,Functional excipients: NaHCO_3_, Na_2_CO_3_, MgO, Mg(OH)_2_,Obtained by saturated aqueous solution method[29]In vitroMicrocapsules composed of Lactobacillus acidophilus surface layer protein,Functional excipients: ATCC 4356 S-layer protein[224]In vitroComplexes of latex particles with API,Functional polymers: Aquateric (cellulose acetophthalate latex),Obtained by adsorption method[228]In vitroImmediate-release microparticles,Functional polymers: Kollicoat IR, hydroxypropyl-β-cyclodextrin,Obtained by spray-drying or freeze-drying[233]In vitroGastro-resistant microparticles,Functional polymers: Eudragit S 100, hydroxypropyl-β-cyclodextrin (carrier),Obtained by spray-drying or emulsification method[234]Omeprazole + piperineIn vivo PK and bioavailability studies (rabbits)Gastroretentive microspheres,Functional polymers: ethylcellulose, HPMCObtained by emulsification-solvent evaporation method[221]Omeprazole + clarithromycinIn vitroSustained-release mucoadhesive microspheres,Functional polymers: HPMC K4M/K100M, Carbopol 971pObtained by non-aqueous emulsification-solvent evaporation method[208]PantoprazoleIn vitroGastro-resistant microparticles with improved photostability,Functional polymers: Eudragit S 100,Obtained by solvent evaporation method[51]In vitroMicroparticles with improved photostability,Functional polymers: Eudragit S 100, poly(e-caprolactone), HPMC,Obtained by emulsification- solvent evaporation method or spray-drying[52]In vitroGastro-resistant microparticles,Functional polymers: Eudragit S 100, HPMCP (HP–55),Obtained by emulsion-solvent evaporation method[200]In vitroSustained-release floating microspheres,Functional polymers: Eudragit S 100, HPMC K100M,Obtained by non-aqueous solvent evaporation method[202]In vitroDouble-walled sustained-release microspheres,Functional polymers: HPMC, sodium alginate (1st layer), Eudragit RS 100 (2nd layer),Obtained by emulsification- solvent evaporation method[205]In vitroSustained-release microsponges,Functional polymers: Eudragit RS 100,Obtained by quasi-emulsion solvent diffusion method[206]In vitroIn vivo antiulcer activity (rats)Enteric-coated, controlled-release microparticles,Functional polymers: Eudragit S 100, poly(ε-caprolactone),Obtained by solvent evaporation method[209]In vitroIn vivo antiulcer activity (rats)Gastro-resistant microparticles,Functional polymers: Eudragit S 100,Obtained by O/O emulsification- solvent evaporation method[210]In vitroIn vivo antiulcer activity (rats)Floating microballons,Functional polymers: Eudragit L 100, Eudragit RS 100,Obtained by emulsion solvent diffusion method[211]In vitroIn vivo antiulcer activity (rats)Gastro-resistant, controlled-release microparticles,Functional polymers: Eudragit S 100, Methocel F4MObtained by spray-drying[215]In vitroGastro-resistant microparticles (agglomerates),Functional polymers: Eudragit S 100, Methocel F4M,Obtained by spray-drying[216]In vitroGastro-resistant microparticles,Functional polymers: Eudragit S 100,Obtained by spray-drying (performed in various conditions)[218]In vivo bioavailability study (dogs)Gastro-resistant microparticles (soft agglomerates),Functional polymers: Eudragit S 100,Obtained by spray-drying[219]In vitroIn vivo antiulcer activity (rats)Gastro-resistant microparticles,Functional polymers: Eudragit S 100, Eudragit RS 100,Obtained by spray-drying[222]In vitroGastro-resistant microspheres,Functional polymers: ethylcellulose, HPC (1st layer), Eudragit L-100, sodium alginate (2nd layer),Obtained by emulsification-solvent evaporation method[223]LansoprazoleIn vitroIn vivo PK and antiulcer activity studies (rats)Enteric-coated, sustained-release microparticles,Functional polymers: Eudragit RS 100, Eudragit S 100, HPMCP (HP–55),Obtained by solvent evaporation method, coated in fluidized bed[199]In vitroEnteric-coated microspheres,Functional polymers: cellulose acetate phthalate (CAP),Obtained by solvent evaporation method[204]In vitroSustained-release, floating microspheres,Functional polymers: ethylcellulose, HPMC,Obtained by solvent evaporation method[207]In vitroEnteric-coated micropellets,Functional polymers: HPMC E5 (sublayer), Acrycoat L-30DObtained by fluid bed coating[212]In vitroSustained-release, floating micropellets,Functional polymers: HPMC, MC, chitosan,Obtained by emulsion- solvent diffusion method[213]In vitroGastro-resistant microparticles,Functional polymers: Eudragit S 100, Eudragit L 100, Eudragit L100-55,Obtained by spray-drying[217]In vitroEnteric-coated, sustained-release microspheres,Functional polymers: Eudragit RS 100 (1st layer), HPMCP (HP-55) (2nd layer),Obtained by solvent evaporation method and spray-drying[220]In vitroCyclodextrin metal-organic frameworks (CD-MOFs) microparticles with improved thermostability,Functional excipients: γ-CDs, KOH, cetyltrimethyl ammonium bromide (CTAB) (stabilizer)[227]RabeprazoleIn vitroIn vivo antiulcer activity (rats)Gastro-resistant, sustained-release mucoadhesive microspheres,Functional polymers: ethylcellulose, Eudragit L 100, HPMC, CMC sodium, HEC, HPC,Obtained by solvent evaporation method and dip coating technique[198]In vitroSustained-release floating microspheres,Functional polymers: ethylcellulose, HPMC K15M,Obtained by emulsification- solvent evaporation method[201]In vitroIn vivo floating study (rabbits)Controlled-release floating microbeads,Functional polymers: sodium alginate, HPMC, BaCl_2_/ CaCl_2_ (crosslinking agents),Obtained by ionotropic gelation method[235]Rabeprazole + amoxicillinIn vitroIn vivo antiulcer activity and radiographic study (rats)Sustained-release microballoons,Functional polymers: Eudragit S 100, HPMC,Obtained by emulsion solvent diffusion method[214]EsomeprazoleIn vitroSustained-release floating microspheres,Functional polymers: HPMC, MC, chitosan,Obtained by solvent evaporation method[203]MinitabletsOmeprazoleIn vitroEnteric-coated minitablets,Functional coating: HPMC (sublayer), Eudragit L 30D-55,Obtained by direct compression, coated in fluidized bed[236]PantoprazoleIn vitroEnteric-coated minitablets,Functional coating: Eudragit L 30D-55, Acryl-Eze IIObtained by direct compression, coated in fluidized bed[237]PelletsOmeprazoleIn vitroIn vivo PK and gastro-resistance studies (dogs/rats)Enteric-coated pellets,Functional polymers: HPMC (sublayer), Eudragit L 30D-55,Core pellets coated in fluidized bed[239]In vitroIn vivo PK and bioequivalence studies (rabbits)Delayed-release pellets,Functional excipients: MMC, lactose, PVP K30,Obtained by sieving-spheronization and extrusion-spheronization methods[242]In vitroIn silico (ANN, modelling tablet properties)Enteric-coated pellets,Functional polymers: HPMC (sublayer), Eudragit L 30D-55,Core pellets coated in fluidized bed[248]In vitroGastro-resistant, alginate beads,Functional polymers: sodium alginate, SBA-15 mesoporous matrix[251]In vitroMultiparticulate pulsatile drug delivery system,Functional excipients: HPMC, ethylcellulose, Eudragit RS 30D, Eudragit RL 30D, NaCl (osmogent),Obtained by film casting/extrusion-spheronization method or fluid bed coating[252]PantoprazoleIn vitroEnteric-coated pellets,Functional polymers: Eudragit L100-55 (organic/aqueous dispersion), Eudragit L 30D-55,Obtained by film casting and extrusion-spheronization method[241]LansoprazoleIn vitroGastro-resistant multilayer pellets,Functional excipients: Na_2_CO_3_ (alkaline layer), HPMC (sublayer), Eudragit L 30D-55 (outerlayer),Core pellets coated in fluidized bed[31]In vitroEnteric-coated pellets,Functional polymers: HPMC (sublayer), Eudragit L100-55,Coated in fluidized bed[238]In vitroGastro-resistant pellets,Functional excipients: carboxymethyl tamarind kernel powder (CMTKP), croscarmellose sodium, MCC,Obtained by extrusion-spheronization method[240]In vitroIn vivo PK study (dogs)Gastro-resistant pellets,Functional polymers: HPMC, aqueous enteric coating,Obtained by fluid-bed granulation, coated in fluidized bed[243]In vitroEnteric-coated pellets,Functional polymers: HPMC (sublayer), methacrylic copolymer,Coated in fluidized bed[247]In vitroIn vivo bioavailability study (dogs)Enteric-coated pellets,Functional polymers: HPMC (sublayer), Eudragit L dispersion, HPMCAS,Coated in fluidized bed[249]RabeprazoleIn vitroEnteric-coated pellets,Functional coating: HPMC (sublayer), Eudragit L 30D-55,Coated in fluidized bed[245]In vitroEnteric-coated pellets,Functional coating: HPMC (sublayer), Eudragit L 30D-55,Coated in fluidized bed[246]In vitroEnteric-coated pellets,Functional coating: Eudragit L 30D-55 or HPMCP (HP–55),Coated in fluidized bed[250]EsomeprazoleIn vitroIn vivo PK study (rats)IVIVCSustained-release, enteric-coated pellets,Functional coating: HPC-EF/HPMC-E5 (sublayer), Eudragit RS 30D/RL 30D (1st layer), Eudragit L 30D-55 (2nd layer),Coated in fluidized bed[30]In vitroIn silico (ANN, coating process)Enteric-coated pellets,Functional coating: HPMC (sublayer), Eudragit L 30D-55,Coated in fluidized bed[244]TabletsOmeprazoleIn vitroEnteric-coated tablets[263]In vitroEnteric-coated tablets,Functional coating: HPMCP, Eudragit S 100 or CAP plasticized with dibutyl phthalate[264]In vitroLyophilized orally disintegrating tablets containing enteric-coated pellets,Fast disintegration of tablets combined with gastric resistance of pellets and their immediate release in phosphate buffer[286]Omeprazole + domperidoneIn vitroDirectly compressed fast disintegrating tablets,Combination of two APIs in one tablet,No stability considerations[287]PantoprazoleIn vitroAPI complex with rosin used to protect it from low pH,Complexes directly tabletted with different superdisintegrants: sodium starch glycolate, crospovidone, croscarmellose sodium[262]In vitroIn vivo antiulcer activity (rats)Sustained-release, enteric-coated tablets,Slow release was achieved by forming matrix using HPMC, cassava starch or PVP,Enteric coating: cellulose acetate phthalate (CAP) or Eudragit L 100,No degradation of API in acid phase detected; prolonged release for 10 h in a buffer stage (first-order kinetic)[269]In vitroPulsatile drug delivery system,Immediate-release tablets press-coated with ethylcellulose and HPMC mixed in different ratios,Drug release lag time from 1.5 up to 3 h was achieved[273]In vitroMultiunit particulate system tablets (MUPS),Pantoprazole pellets coated with Eudragit L and with cushion layer,Fast-disintegrating tablets were achieved with a drug release in the acid phase lower than 6%, followed by immediate release in the buffer[274]In vitroOrodispersible tablets with crospovidone or sodium starch glycolate were directly compressed with API,Stability issues were not considered[288]LansoprazoleIn vitroDelayed-release tablets,Immediate-release tablets press-coated with ethylcellulose and two different grades of HPMC,Drug release lag time from 2 up to 4 h was achieved[257]In vitroHot-melt extrusion used to combine lansoprazole with PVP, Lutrol F68 and magnesium oxide,Extrudates compressed to core tablets, which were coated with Eudragit L100-55,No drug release during 1 h acid stage, followed by immediate release to a buffer[266]In vitroIn vivo absorption studies (dogs), disintegration time in the mouth (human)Orodispersible tablets containing enteric coated microgranules,Enteric coating: Eudragit L30D-55 and Eudragit NE30D,Bioequivalence was demonstrated with a manufactured drug[276,277,278]In vitroFast-dissolving tablets,For solubility improvement solid dispersions of lansoprazole with PEG 4000/6000 or drug-β-cyclodextrin complexes were formed,Tablets prepared by direct compression with superdisintegrants,Degradation of API in acid was not considered in the study[279]In vivo (human)Clinical trialsOrodispersible tablets,A review of studies on the clinical effectiveness of orodispersible tablets with lansoprazole[280,281]In vivo (human) In the study the effect of water intake on lansoprazole absorption from orodispersible tablets was evaluated,No significant difference between administration with or without water was observed[282]In vivo bioequivalence studies (human)Bioequivalence studies on orodispersible tablets and capsules containing lansoprazole,No significant differences between C_max_ and AUC values of tested formulations were observed[283]In vitroIn vivo (human)Comparison of branded and five generic orodispersible tablets containing lansoprazole,Formulation quality (stability in saliva, dissolution in acidic and intestinal media) and ingestibility were tested[284]In vitroStudy on physical properties of six different orodispersible tablets containing lansoprazole[285]RabeprazoleIn vitroIn vivo PK studies (beagle dogs)Immediate-release formulation containing rabeprazole core tablet dry-coated with sodium bicarbonate,Faster onset of action in comparison to reference tablets[33]In vitroEnteric-coated tablets,DrugCoat L100 (anionic copolymer based on methacrylic acid and ethyl acrylate) used as a coating polymer[253]In vitroEnteric-coated tablets,Core tablets prepared by direct compression or after wet granulation,Coating with HPMCP (Instacoat EN-HPMCP)[258]In vitroEnteric-coated tablets,Coating with HPMCP,Drug release in buffer stage extended up to 12 h[259]In vitroSustained-release tablets,Matrix tablet were prepared after wet granulation of API with HPMC, Carbopol or sodium carboxymethyl cellulose,Degradation of API in acid was not considered in the study[270]In vitroOrodispersible tablets,Different superdisintegrants evaluated: crospovidone, croscarmellose sodium, pregelatinized starch, L-HPC, treated agar ,Degradation of API in acid was not considered in the study[289]EsomeprazoleIn vitroEnteric-coated tablets ,Coating with Eudragit L 30D-55, HPMCP, CAP or Acryl-EZE[256]In vitroEnteric-coated tablets ,Core tablets prepared after wet granulation,Coating with Instacoat EN-Super-II[260]In vitroEx vivo permeation studies (porcine mucosa)In vivo pharmacokinetics studies (rats)Immediate-release tablets containing magnesium oxide or sodium bicarbonate as an acid protective ingredients,Minitablets with esomeprazole and sodium bicarbonate coated with Eudragit L100-55,Addition of bicarbonate promoted esomeprazole permeation and its immediate absorption[261]In vitroColon-specific drug delivery system,Core tablets were press-coated with a mixture of HPMCP and ethyl cellulose,Drug release sustained up to 6 h in buffer stage[267]In vitroMultiunit particulate system (MUPS) tablets,Enteric-coated pellets compressed with different excipients (lactose, dibasic calcium phosphate) to form of tablet,High resistance to acid degradation, followed by immediate API release in buffer[275]DexlansoprazoleIn vitroExtended-release tablets,Directly compressed tablets containing HPMC and HPMCP were coated with shellac,Drug release extended up to 12 h,[272]TenatoprazoleIn vitroEnteric-coated tablets,Directly compressed tablets were coated with HPMCP, Eudragit L 30D-55, or HPMCAS,Drug release extended up to 12 h[255]In vitroExtended-release matrix tablets,Direct compression of API with polymers such as Carbopol, Methocel or Eudragit, and sodium bicarbonate as a pH controlling agent,Drug release extended up to 12 h[268]IlaprazoleIn vitroEnteric-coated tablets,Compression of the core tablets after wet granulation,Coating with Eudragit L 100 or HPMCP,Efficient gastric protection followed by immediate release in buffer stage was achieved[254]

In vitroExtended-release tablets,Direct compression of the core tablets containing different superdisintegrants,Coating with HPMCP and Eudragit L 100,Efficient gastric protection followed by drug release extended up to 12 h was achieved[271]Fixed-dose combination productsEsomeprazole + naproxenIn vitroHot-melt co-extrusion was used to produce cylindrical systems,The core of the cylinder contained naproxen with enteric polymers like Eudragit, HPMC-AS-LF, HPMCP or Eudragit L100-55,The outer layer of the cylinder contained esomeprazole with immediate release polymers such as Kollidon, Klucel, Methocel or PEO,Degradation of esomeprazole in acidic medium was not considered in the study[290]Bilayer tabletsLansoprazole + amoxycillinIn vitroBilayer tablets,Immediate-release layer containing lansoprazole, sodium starch glycolate and MCC,Sustained release layer with amoxicillin, HPMC and EC,Degradation of lansoprazole in acidic medium was not considered in the study[293]Esomeprazole + aceclofenacIn vitroBilayer floating tablets,Immediate-release layer contained esomeprazole, sodium bicarbonate, citric acid, and sodium starch glycolate,Sustained release layer contained HPMC in different grades and xanthan gum,Degradation of esomeprazole in acidic medium was not considered in the study[294]Esomeprazole + clarithromycinIn vitroControlled-release floating effervescent bilayer tablets,Combination of Eudragit RS 100 and Carbopol was used to control drug release in both layers,Sustained release for up to 24 h was achieved,Degradation of esomeprazole in acidic medium was not considered in the study[291]Esomeprazole + levosulpirideIn vitroBilayer tablets,Immediate-release layer containing esomeprazole with superdisintegrants such as croscarmellose sodium, crospovidone or sodium starch glycolate,Sustained release floating layer contained levosulpiride, HPMC, sodium bicarbonate and citric acid,Immediate release of esomeprazole and 12 h release of levosulpiride were achieved,Degradation of esomeprazole in acidic medium was not considered in the study[292]Floating tabletsPantoprazoleIn vitroFloating effervescent tablets,Tablets contained pectin, HPMC, sodium bicarbonate, calcium carbonate and citric acid granulated with isopropyl alcohol prior to compression,Pantoprazole release was extended up to 8 h,Degradation of pantoprazole in acidic medium was not considered in the study[295]In vitroSustained release floating tablets,Direct compression used to prepare tablets containing API, HPMC or sodium alginate with MCC and sodium bicarbonate,Pantoprazole release was extended up to 8 h,Degradation of pantoprazole in acidic medium was not considered in the study[297]LansoprazoleIn vitroSustained release floating tablets,Direct compression used to prepare tablets containing API, HPMC and sodium bicarbonate,Lansoprazole release extended to 10 h,Degradation of API in acidic medium was not considered in the study[296]In vitroSustained release effervescent floating tablets,Direct compression used to prepare tablets containing API, xanthan gum, gellan gum, Carbopol or chitosan, citric acid and sodium bicarbonate,Lansoprazole release extended to 12 h,Degradation of API in acidic medium was not considered in the study[299]RabeprazoleIn vitroIn vivo pharmacokinetic and antiulcer activity studies (rats)Immediate-release floating tablets,Wet granulation with ethanolic solution of HPMC was used to prepare granules containing API, pectin, mannitol, PEG 400, sodium bicarbonate, calcium carbonate and citric acid,Floating tablets neutralize gastric acid to protect API from degradation,Faster onset of action and better antiulcer activity was achieved as compared to the commercial rabeprazole delayed-release capsules[298]Hydrogel formulationsPantoprazoleIn vitroColon-specific controlled release hydrogel,Gum tragacanth and acrylic acid based hydrogel was prepared by graft copolymerization,pH-sensitive drug release rate was achieved,Pantoprazole released extended up to 30 h[301]In vitroSuperporous hydrogel with pantoprazole,Methacrylic acid and acrylamide were polymerized in the presence of N,N-methylene-bis-acrylamide as crosslinking agent,High acid resistance and extended pantoprazole release up to 6 h was achieved[302]In vitroIn vivo studies on hydrogel gastro-retention (mice)In situ gelling formulation,Gellan gum, sodium alginate and HPMC were used as a gelling agents,Pantoprazole release extended to 12 h,Degradation of API in acidic medium was not considered in the study[303]RabeprazoleIn vitroHydrogel beads intended for the colon delivery of rabeprazole,Hydrogel beads were prepared by ionotropic gelation of sodium alginate with calcium chloride,Eudragit S100 used for enteric-coating of beads,Gastric protection of rabeprazole was achieved followed by 8 h drug release[304]Mucoadhesive tabletsOmeprazoleIn vitroMucoadhesive tablets with pellets inside,Tablets were coated with mucoadhesive polymer: HPMC K4M, sodium carboxymethylcellulose, ethyl cellulose or Carbopol 934P, and then with Eudragit L100 to achieve final enteric coating[305]In vitroIn vivo studies on absorption from the oral cavity and tablets adhesion to the oral mucosa (human)Buccal adhesive tablets,Sodium alginate and HPMC were used as a mucoadhesive polymers,Magnesium oxide, potassium phosphate monobasic, sodium phosphate monobasic and dibasic were used as a pH-stabilizers,Stability of API in human saliva for 4 h was achieved[307]In vitroIn vivo pharmacokinetic studies (hamsters)Omeprazole buccal adhesive tablets,API was directly compressed with sodium alginate, HPMC, magnesium oxide, and croscarmellose sodium,Sustained drug release was confirmed in pharmacokinetic studies; constant omeprazole level in blood was maintained for 6 h[308]In vitroIn vivo pharmacokinetic studies (hamster), mucoadhesive force measurement (human)Buccal adhesive tablets with omeprazole,API was directly compressed with sodium alginate, HPMC, magnesium oxide, and croscarmellose sodium,Sustained drug release was confirmed in pharmacokinetic studies; constant omeprazole level in blood was maintained for 6 h[309]In vitroPediatric buccal film,Casting method was used to prepare films with omeprazole, HPMC, MC, sodium alginate, carrageenan, metolose, PEG 400 and L-arginine[310]PantoprazoleIn vitroSustained release mucoadhesive gastroretentive system,Tablets containing API, MCC, PVP, and HPMC, Carbopol, or guar gum were prepared with direct compression,Extended release of pantoprazole was achieved for 10 h,Degradation of API was not analyzed in the study[306]Oral liquid suspensionsOmeprazoleIn vitroStudy on physicochemical and microbiological stability,Suspension composed of crushed omeprazole pellets or pure omeprazole in a complex vehicle[311]In vitroEnteric-coated particles for suspension in syrup (extemporaneously),Functional polymes: Eudragit E 100, Eudragit L100-55,Particles obtained by fluid bed coating[312]In vitroIn vivo preliminary toxicity and antiulcer activity studies (mice)Enteric-coated nanoparticles for oral liquid suspension,Functional polymes: Eudragit RS 100 (1st layer), Eudragit L100-55 (2nd layer),Obtained by interfacial deposition of the preformed polymers method[313]Transdermal deliveryOmeprazoleIn vivo PK study (human)Study on omeprazole transdermal absorption,Transdermal gel formulation: pleuronic lecithin organogel (PLO) containing omeprazole (50 mg/mL)[316]LansoprazoleEx vivo penetration study (pigs)In vivo PK study (rats)Nanostructured lipid carriers (NLCs) for hydrogel formulations,Functional excipients: glyceryl monostearate, stearylamine, SDS, isopropyl myristate, menthol[315]RabeprazoleEx vivo penetration study (snake)Transdermal patches,Film forming polymers: HPC-EF, PVP K30, PVP K90,Obtained by solvent casting method[314]SuppositoriesOmeprazoleIn vitroPediatric suppository,Functional excipients: witepsol H15, arginine (L) base[317]Clinical trial (efficacy, PK)Study on efficacy and pharmacokinetics of omeprazole administered in form of suppositories in infants[318]Intravenous formulationsOmeprazoleIn vitroPowder for solution for infusion with cyclodextrins as stability enhancers,Obtained by lyophilization[319]In vitroNanosuspension,Suspension components: 8.4% sodium bicarbonate solution, Poloxamer 188 (1%),Obtained by DissoCubes^®^ technology[320]


## 5. Conclusions and Future Perspectives

Despite more than 40 years of application of proton pump inhibitors in the pharmacotherapy of many gastrointestinal disorders, there are still multiple unresolved issues that needs to be met to ensure the stability, efficacy and above all safety of the application of PPIs. As this review shows, there are many approaches to alleviate PPIs’ disadvantages, but there is still room for improvement when both patient experience and therapeutic efficacy and safety are at stake.

Probably one of the most important issues is the availability of PPIs to children of all ages and medical conditions. Although there are some dosage forms with PPIs intended for children, there is still no universal one, which might be convenient for all pediatric groups. This induces the common need to open capsules, withdraw their content, sometimes suspend it in a random liquid, or adjust the dose. Moreover, it is common practice in many countries to crush the pellets as well as to prepare their dispersion in alkali media such as sodium bicarbonate. It may partially protect the drug substance, but may not be sufficient to provide patients with an optimal dose delivered to the intestines where PPIs are absorbed. Therefore, there is still a strong demand to develop formulations that balance the bioavailability of PPIs with their stability and taste-masking properties. There seems to be a great deal of hope in the incorporation of micro- and nanoparticles into orodispersible tablets (ODTs), minitablets (MODTs), films (ODFs), or granules. However, this is still an unexplored field in the case of PPIs and needs to be investigated more deeply.

Another great opportunity is the design and development of more stable PPIs such as AGN 201904-Z, which is actually a prodrug converted in the systemic circulation to omeprazole. It is acid stable and therefore does not require an enteric coating or other protection from the acidic environment in the stomach. However, as a new drug moiety, it is still necessary to prove its safety and efficacy in a larger group of patients, including children and the elderly [21,22,23].

## Figures and Tables

**Figure 1 pharmaceutics-14-02043-f001:**
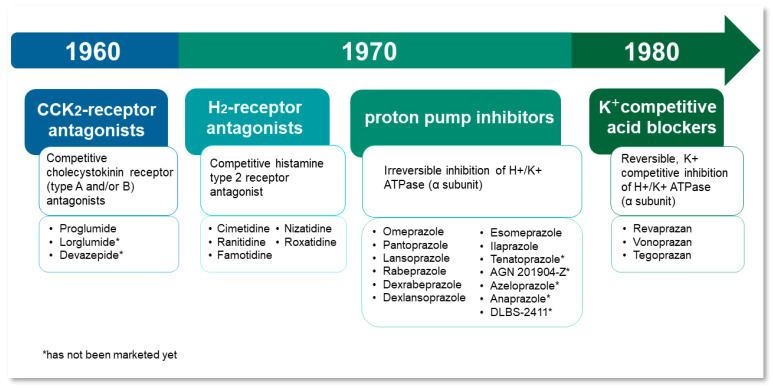
Antisecretory drugs [5,6,9,10,11,13,14,15,16,17,18].

**Figure 2 pharmaceutics-14-02043-f002:**
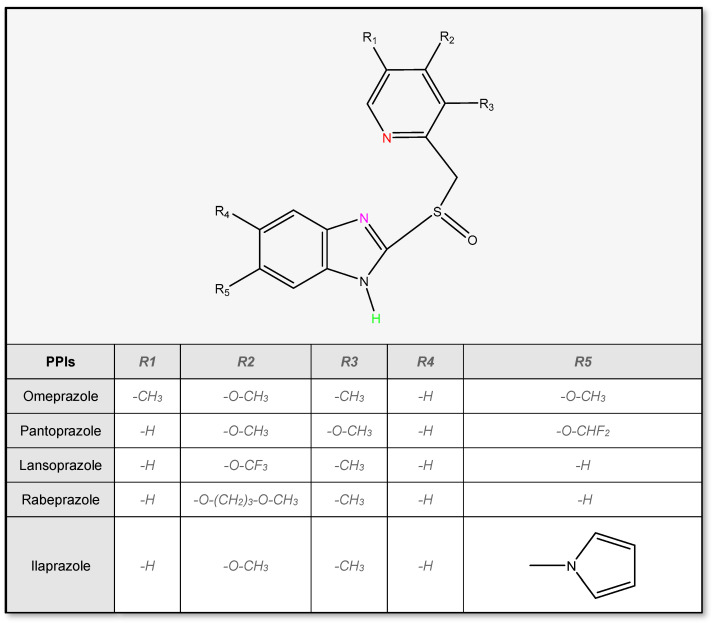
Chemical structure of proton pump inhibitors.

**Figure 3 pharmaceutics-14-02043-f003:**
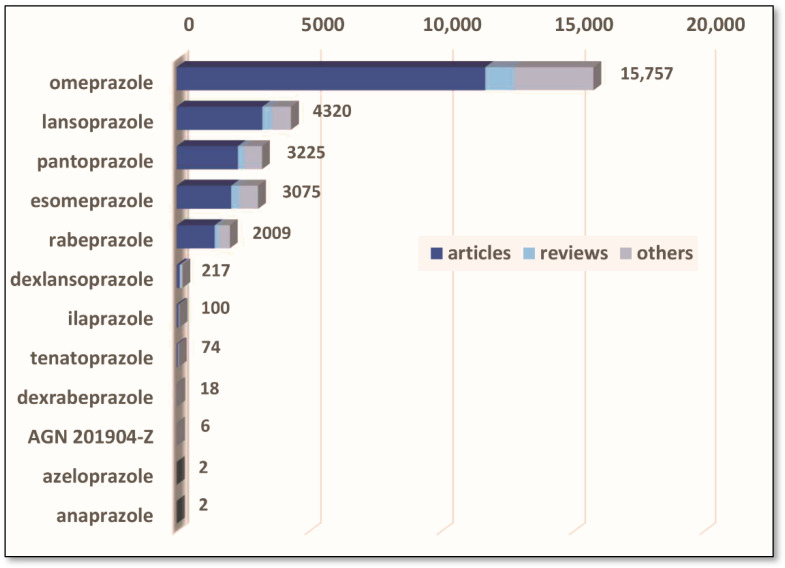
Number of articles, and other items published on proton pump inhibitors in the years 1990–2022 (source: Web of Science Core Collection).

**Table 1 pharmaceutics-14-02043-t001:** Pharmacological characteristics and medical uses of Proton Pump Inhibitors (according to a DrugBank [20,26,28,29,30,31,32,33,34]).

PPIs	pKa	logP	Solubility [mg/mL]	BCS Classification	Half-Life [h]	t_max_ [h]	C_max_ [μmol/L]
Omeprazole	4.77 9.29	1.66	0.359	II	0.5–1	0.5–3.5	0.23–23.2 (20 mg)
Pantoprazole	3.55 9.15	2.11	0.495	III	1–1.9	2–3	2.87–8.61 (40 mg)
Lansoprazole	4.16 9.35	2.84	0.250	II	1.6	1.7	1.62–3.25 (30 mg)
Rabeprazole	4.24 9.35	2.04	0.336	III	1–2	2–5	1.14 (20 mg)
Esomeprazole	4.77 9.68	1.66	0.353	II	1–1.5	1.5	2.1–2.4 (20 mg)
Dexlansoprazole	4.16 9.35	2.84	0.250	II	1–2	1–6	1.87 (30 mg)
Ilaprazole	4.27 10.10	2.42	0.0934	N/A	3.0–3.4	0.75–1.0	4.2–5.1 (20 mg)

**Table 4 pharmaceutics-14-02043-t004:** Examples of proton pump inhibitors in fixed-dose combination products (available in the U.S. and/or EEA/UK).

APIs	Dosage	Drug Formulation	Brand Name	References
Omeprazole + Diclofenac	20 mg + 75 mg	Capsule with modified release pellets	DicloDuo Combi Diclopram	[88,174,176]
Omeprazole sodium + Macrogol 400	40 mg	Powder and solvent for solution for injection	Omeprazole Sandoz	[88,161]
Amoxicillin + Lansoprazole + Metronidazole	500 mg + 30 mg + 400 mg	Combination pack (film coated tablet + GR capsule + tablet)	Helipak A	[88,177]
Amoxicillin Trihydrate + Clarithromycin + Omeprazole	500 mg + 500 mg + 30 mg	Combination pack (film coated tablet + film coated tablet + GR capsule)	Helipak K	[88,178]
Amoxicillin + Clarithromycin + Pantoprazole	1000 mg + 500 mg +40 mg	Combination pack (film coated tablet + film coated tablet + GR tablet)	Panclamox Zacpac	[88,175]
Omeprazole + Sodium bicarbonate	20 mg + 1.68 g 40 mg + 1.68 g	Oral suspension	Zegerid	[87]
	20 mg + 1.1 g 40 mg + 1.1 g	Capsule		
Omeprazole magnesium + Amoxicilin + Rifabutin	10 mg + 250 mg + 12.5 mg	DR capsule	Talicia	
Lansoprazole + Amoxicillin + Clarithromycin	30 mg + 500 mg + 500 mg	Combination pack (DR capsule + capsule + tablet)	Prevpac	
Esomeprazole magnesium + Naproxen	20 mg + 375 mg	GR tablet	Vimovo	[87,173,179]
	20 mg + 500 mg			

## Data Availability

Not applicable.

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
