# Peer review of "Formulation of Dosage Forms with Proton Pump Inhibitors: State of the Art, Challenges and Future Perspectives"

_pharmaceutics, 2022, doi:10.3390/pharmaceutics14102043_

Round 1

Reviewer 1 Report

In this manuscript, the authors provided a review on the state of the art, challenges and future perspective of formulation of dosage forms with PPIs.

The authors clearly did a thorough job coming through the literature to produce this review which is packed with information. The only comment I have for this manuscript is to please have someone read it for English.

Reviewer 2 Report

Review comments on pharmaceutics-1917237: Formulation of dosage forms with proton pump inhibitors: state of the art, challenges and future perspectives

In this manuscript, the authors summarized the historical background, PK, and PD of PPIs, challenges in drug formulations, and dosage forms available on the market. The major part of this review manuscript is the overview of the recent development of new PPI formulations. The authors put much effort into collecting data and preparing the manuscript. Overall, this manuscript is of high interest to readers in Pharmaceutics. However, there are several issues in this manuscript. Prior to publication, the authors should improve the manuscript by considering the comments below.

1. In the Abstract and Introduction section, the authors should highlight the novelty and contribution of this review. The authors only mentioned that “the development of novel improved medicinal products with PPIs is fully reasonable and ought to be supported”. However, this manuscript summarized the recent development of new PPI formulations. The authors should signify that various approaches have been employed to prepare new PPI formulations, and due to that, this review is conducted.

2. Section 2 is “challenges in drug formulation”. However, some sub-sections are not suitable to put in this section, including “Influence of enteric polymers on PPIs stability” (lines 318-392) [should be moved to formulation approaches] and “Analytical methods for PPIs determination” (lines 394-449) [should be in another section]. The “Analytical methods for PPIs determination” should be separated into small parts, and more data on LC-MS/MS methods should be added.

3. Section 4: many studies presented in this section seemed not to have PK data. How about the clinical status of these formulations?

4. Information in Table 5 was insufficient. This table can be omitted, or more information should be added (e.g., short descriptions of formulation approaches, in vitro and in vivo outcomes). 

Reviewer 3 Report

Proton pump inhibitors (PPIs) are a class of drugs used to reduce stomach acid and relieve GERD symptoms. The manuscript provides a comprehensive review on the formulation of PPIs. It introduced the history of the development of PPIs, summarized the available formulations of PPIs, identified the potential challenges, and pointed out the possible avenues of future research.

This manuscript is well-written and structured properly. The tables and figure are formatted properly. Sufficient references are categorized using tables and summarized in the end.

Round 2

Reviewer 2 Report

The manuscript was appropriately revised and can be accepted as is.